# PINNACLE: PINN ADAPTIVE COLLOCATION AND EXPERIMENTAL POINTS SELECTION

**Gregory Kang Ruey Lau**[*†‡]**, Apivich Hemachandra**[*†]**,**
[†]Department of Computer Science, National University of Singapore, Singapore 117417
[‡]CNRS@CREATE, 1 Create Way, #08-01 Create Tower, Singapore 138602
{greglau,apivich}@comp.nus.edu.sg

**See-Kiong Ng & Bryan Kian Hsiang Low**
Department of Computer Science, National University of Singapore, Singapore 117417
seekiong@nus.edu.sg, lowkh@comp.nus.edu.sg

## ABSTRACT

Physics-Informed Neural Networks (PINNs), which incorporate PDEs as soft constraints, train with a composite loss function that contains multiple training point types: different types of *collocation points* chosen during training to enforce each PDE and initial/boundary conditions, and *experimental points* which are usually costly to obtain via experiments or simulations. Training PINNs using this loss function is challenging as it typically requires selecting large numbers of points of different types, each with different training dynamics. Unlike past works that focused on the selection of either collocation or experimental points, this work introduces PINN ADAPTIVE COLLOCATION AND EXPERIMENTAL POINTS SELECTION (PINNACLE), the first algorithm that *jointly optimizes the selection of all training point types, while automatically adjusting the proportion of collocation point types as training progresses*. PINNACLE uses information on the interaction among training point types, which had not been considered before, based on an analysis of PINN training dynamics via the Neural Tangent Kernel (NTK). We theoretically show that the criterion used by PINNACLE is related to the PINN generalization error, and empirically demonstrate that PINNACLE is able to outperform existing point selection methods for forward, inverse, and transfer learning problems.

## 1 INTRODUCTION

Deep learning (DL) successes in domains with massive datasets have led to questions on whether it can also be efficiently applied to the scientific domains. In these settings, while training data may be more limited, domain knowledge could compensate by serving as inductive biases for DL training. Such knowledge can take the form of governing Partial Differential Equations (PDEs), which can describe phenomena such as conservation laws or dynamic system evolution in areas such as fluid dynamics (Cai et al., 2021; Chen et al., 2021; Jagtap et al., 2022), wave propagation and optics (bin Waheed et al., 2021; Lin & Chen, 2022), or epidemiology (Rodríguez et al., 2023).

Physics-Informed Neural Networks (PINNs) are neural networks that incorporate PDEs and their initial/boundary conditions (IC/BCs) as soft constraints during training (Raissi et al., 2019), and have been successfully applied to various problems. These include forward problems (i.e., predicting PDE solutions given specified PDEs and ICs/BCs) and inverse problems (i.e., learning unknown PDE parameters given experimental data). However, the training of PINNs is challenging.

To learn solutions that respect the PDE and IC/BC constraints, PINNs need a composite loss function that requires multiple training point types: different types of *collocation points* (CL points) chosen during training to enforce each PDE and IC/BCs, and *experimental points* (EXP points) obtained via experiments or simulations that are queries for ground truth solution values. These points have

---

[*]Equal contribution.

different training dynamics, making training difficult especially in problems with complex structure. In practice, getting accurate results from PINNs also require large numbers of CL and EXP points, both which lead to high costs – the former leads to high computational costs during the training process (Cho et al., 2023; Chiu et al., 2022), while the latter is generally costly to acquire experimentally or simulate. Past works have tried to separately address these individually by considering an adaptive selection of CL points (Nabian et al., 2021; Gao & Wang, 2023; Wu et al., 2023; Peng et al., 2022; Zeng et al., 2022; Tang et al., 2023), or EXP points selection through traditional active learning methods (Jiang et al., 2022; Cuomo et al., 2022). Some works have also proposed heuristics that adjust loss term weights of the various point types to try improve training dynamics, but do not consider point selection (Wang et al., 2022c). However, no work thus far has looked into optimizing all training point types jointly to significantly boost PINN performance.

Given that the solution spaces of the PDE, IC/BC and underlying output function are tightly coupled, it is inefficient and sub-optimal to select each type of training points separately and ignore cross information across point types. For different PINNs or even during different stages of training, some types of training points may also be less important than others.

In this paper, we introduce the algorithm PINN ADAPTIVE COLLOCATION AND EXPERIMENTAL DATA SELECTION (PINNACLE) that is the first to jointly optimize the selection of all types of training points (e.g., PDE and IC/BC CL and EXP points) given a budget, *making use of cross information among the various types of points* that past works had not considered to provide significant PINNs training improvements. Our contributions are summarized as follows:

- We introduce the problem of *joint point selection* for improving PINNs training (Sec. 3), and propose a novel representation for the augmented input space for PINNs that enables all types of training points for PINNs to be analyzed *simultaneously* (Sec. 4.1).

- With this augmented input space, we analyze the PINNs training dynamics using the combined NTK eigenspectrum which naturally incorporates the cross information among the various point types encoded by the PDEs and IC/BC constraints (Sec. 4.2).

- Based on this analysis, we define a new notion of convergence for PINNs (*convergence degree*) (Sec. 4.3) that characterizes how much a candidate set involving multiple types of training points would help in the training convergence for the entire augmented space. We also theoretically show that selecting training points that maximizes the convergence degree leads to lower generalization error bound for PINNs (Sec. 4.3).

- We present a computation method of the convergence degree using Nystrom approximation (Sec. 5.1), and two variants of PINNACLE based on maximizing the convergence degree while also considering the evolution of the empirical NTK (eNTK) of the PINN (Sec. 5.2).

- We empirically illustrate how PINNACLE's automated point selection across all point types are interpretable and similar to heuristics of past works, and demonstrate how PINNACLE outperform benchmarks for a range of PDEs in various problem settings: forward problems, inverse problems, and transfer learning of PINNs for perturbed ICs (Sec. 6).

## 2 BACKGROUND

**Physics-Informed Neural Networks.** Consider partial differential equations (PDEs) of the form

$$\mathcal{N}[u, \beta](x) = f(x) \quad \forall x \in \mathcal{X} \qquad \text{and} \qquad \mathcal{B}_i[u](x_i') = g_i(x_i') \quad \forall x_i' \in \partial \mathcal{X}_i \qquad (1)$$

where $u(x)$ is the function of interest over a coordinate variable $x$ defined on a bounded domain $\mathcal{X} \subset \mathbb{R}^d$ (where time could be a subcomponent), $\mathcal{N}$ is a PDE operator acting on $u(x)$ with parameters[1] $\beta$, and $\mathcal{B}_i[u]$ are initial/boundary conditions (IC/BCs) for boundaries $\partial \mathcal{X}_i \subset \mathcal{X}$. For PINN training, we assume operators $\mathcal{N}$ and $\mathcal{B}$, and functions $f$ and $g$ are known, while $\beta$ may or may not be known.

Physics-Informed Neural Networks (PINNs) are neural networks (NN) $\hat{u}_\theta$ that approximates $u(x)$. They are trained on a dataset $X$ that can be partitioned into $X_s$, $X_p$ and $X_b$ corresponding to the

---

[1]To simplify notation, we write $\mathcal{N}[u, \beta]$ as $\mathcal{N}[u]$, keeping the $\beta$ dependence implicit, except for cases where the task is to learn $\beta$. Examples of PDEs, specifically those in our experiments, can be found in Appendix J.1.

EXP points, PDE CL points and IC/BC CL points respectively. The PDE and IC/BCs (1) are added as regularization (typically mean-squared errors terms) in the loss function[2]

$$\mathcal{L}(\hat{u}_\theta; X) = \sum_{x \in X_s} \frac{(\hat{u}_\theta(x) - u(x))^2}{2N_s} + \lambda_p \sum_{x \in X_p} \frac{(\mathcal{N}[\hat{u}_\theta](x) - f(x))^2}{2N_p} + \lambda_b \sum_{x \in X_b} \frac{(\mathcal{B}[\hat{u}_\theta](x) - g(x))^2}{2N_b}$$
(2)

where the components penalize the failure of $\hat{u}_\theta$ in satisfying ground truth labels $u(x)$, the PDE, and the IC/BCs constraints respectively, and $\lambda_p$ and $\lambda_b$ are positive scalar weights.

**NTK of PINNs.** The Neural Tangent Kernel (NTK) of PINNs can be expressed as a block matrix broken down into the various loss components in (2) (Wang et al., 2021a). To illustrate, a PINN trained with just the EXP points $X_s$ and PDE CL points $X_p$ using gradient descent (GD) with learning rate $\eta$ has training dynamics that can be described by

$$\begin{bmatrix} \hat{u}_{\theta_{t+1}}(X_s) - \hat{u}_{\theta_t}(X_s) \\ \mathcal{N}[\hat{u}_{\theta_{t+1}}](X_p) - \mathcal{N}[\hat{u}_{\theta_t}](X_p) \end{bmatrix} = -\eta \begin{bmatrix} \Theta_{t,ss} & \Theta_{t,sp} \\ \Theta_{t,ps} & \Theta_{t,pp} \end{bmatrix} \begin{bmatrix} \hat{u}_{\theta_t}(X_s) - u(X_s) \\ \mathcal{N}[\hat{u}_{\theta_t}](X_p) - f(X_p) \end{bmatrix}$$
(3)

where $J_{t,s} = \nabla_\theta \hat{u}_{\theta_t}(X_s)$, $J_{t,p} = \nabla_\theta \mathcal{N}[\hat{u}_{\theta_t}](X_p)$, and $\Theta_{t,ss} = J_{t,s}J_{t,s}^\top$, $\Theta_{t,pp} = J_{t,p}J_{t,p}^\top$, and $\Theta_{t,sp} = \Theta_{t,ps}^\top = J_{t,p}J_{t,s}^\top$, are the submatrices of the PINN empirical NTK (eNTK). Past works have analyzed PINNs training dynamics using the eNTK (Wang et al., 2021a; 2022c; Gao et al., 2023). We provide additional background information on NTKs for general NNs in Appendix D.

## 3 POINT SELECTION PROBLEM SETUP

The goal of PINN training is to minimize the composite loss function (2), which comprises of separate loss terms corresponding to EXP ($X_s \subset \mathcal{X}$), PDE CL ($X_p \subset \mathcal{X}$), and multiple IC/BC CL ($X_b \subset \partial\mathcal{X}$) points[3]. Instead of assuming that the training set is fixed and available without cost, we consider a more realistic problem setting where we have a limited training points budget. The choice of training points then becomes important for good training performance.

Hence, the problem is to select training sets $X_s$, $X_p$, and $X_b$ given a fixed training budget, to achieve the best PINN performance[4]. Due to high acquisition cost (e.g., limited budget for conducting experiments), we consider a fixed EXP training budget $|X_s|$. We also consider a combined training budget for the various CL point types $|X_p| + |X_b| = k$, which is limited due to computational cost at training. Note that the algorithm is allowed to freely allocate the CL budget among the PDE and various IC/BC point types during training, in contrast to other PINNs algorithms where the user needs to manually fix the number of training points for each type. Also, while the EXP and CL points do not share a common budget, they can still be jointly optimized for better performance (i.e., EXP points selection could still benefit from information from CL points and vice versa).

## 4 NTK SPECTRUM AND NN CONVERGENCE IN THE AUGMENTED SPACE

### 4.1 AUGMENTED INPUT REPRESENTATION FOR PINNS

To analyze the interplay among the training dynamics of different point types, we define a new augmented space $\mathcal{Z} \subset \mathcal{X} \times \{s, p, b\}$, containing training points of all types, as

$$\mathcal{Z} \triangleq \{(x, s) : x \in \mathcal{X}\} \cup \{(x, p) : x \in \mathcal{X}\} \cup \{(x, b) : x \in \partial\mathcal{X}\}$$
(4)

where s, p and b are the indicators for the EXP points, PDE CL points and IC/BC CL points respectively, to specify which type of training point $z$ is associated with. Note that a PDE CL point $z = (x, p) \in \mathcal{Z}$ is distinct from an EXP point $z = (x, s)$, even though both points have the same

---

[2]For notational simplicity we will denote a single IC/BC $\mathcal{B}$ in subsequent equations, however the results can easily be generalized to include multiple ICs/BCs as well.

[3]To compare with the active learning problem setting, EXP points can be viewed as queries to the ground truth oracle, and CL points as queries to oracles that advise whether $\hat{u}_\theta$ satisfies the PDE and IC/BC constraints.

[4]The specific metric will depend on the specific setting: for forward problems it will be the overall loss for the function of interest $u(x)$, while for inverse problems it will be for the parameters of interest $\beta$

coordinate $x$. We abuse notation by defining any function $h : \mathcal{X} \to \mathbb{R}$ to also be defined on $\mathcal{Z}$ by $h(z) = h(x, \mathrm{r}) \triangleq h(x)$ for all indicators $\mathrm{r} \in \{\mathrm{s}, \mathrm{p}, \mathrm{b}\}$, and also define a general prediction operator $F$ that applies the appropriate operator depending on the indicator r, i.e.,

$$F[h](x, \mathrm{s}) = h(x), \quad F[h](x, \mathrm{p}) = \mathcal{N}[h](x), \quad \text{and} \quad F[h](x, \mathrm{b}) = \mathcal{B}[h](x). \tag{5}$$

The residual of the network output at any training point $z$ can then be expressed as $R_{\theta_t}(z) = F[\hat{u}_{\theta_t}](z) - F[u](z)$, and we can express the loss (2), given appropriately set $\lambda_r$ and $\lambda_b$, as

$$\mathcal{L}(\hat{u}_\theta; Z) = \frac{1}{2} \sum_{z \in Z} R_{\theta_t}(z)^2 = \frac{1}{2} \sum_{z \in Z} \left( F[\hat{u}_\theta](z) - F[u](z) \right)^2. \tag{6}$$

Using the notations introduced above, the goal of our point selection algorithm would be to select a training set $S$ from $\mathcal{Z}$ which will be the most beneficial for PINNs training. As the point selection is done on $\mathcal{Z}$ space, all types of training points can now be considered together, meaning the algorithm could automatically consider the cross information between each type of points and prioritize the budgets accordingly (explored further in Sec. 6.2).

## 4.2 NTK EIGENBASIS AND TRAINING DYNAMICS INTUITION

With this PINN augmented space, we can now consider different basis that naturally encode cross information on the interactions among the various point types during training. For instance, the off-diagonal term $\Theta_{t,sp}$ in (3) encodes how the PDE residuals, through changes in NN parameters during GD, change the residuals of $\hat{u}_\theta$. This is useful as in practice, we usually have limited EXP data on $\hat{u}_\theta$ but are able to choose PDE CL points more liberally. Past works studying PINN eNTKs (Wang et al., 2021a; 2022c) had only used the diagonal eNTK blocks ($\Theta_{t,ss}, \Theta_{t,pp}$) in their methods.

Specifically, we define the eNTK in the augmented space[5] as $\Theta_t(z, z') = \nabla_\theta F[\hat{u}_{\theta_t}](z) \, \nabla_\theta F[\hat{u}_{\theta_t}](z')^\top$. Since $\Theta_t$ is a continuous, positive semi-definite kernel, by Mercer's theorem (Schölkopf & Smola, 2002) we can decompose $\Theta_t$ to its eigenfunctions $\psi_{t,i}$ and corresponding eigenvalues $\lambda_{t,i} \geq 0$ (which are formally defined in Appendix E) as

$$\Theta_t(z, z') = \sum_{i=1}^{\infty} \lambda_{t,i} \, \psi_{t,i}(z) \psi_{t,i}(z'). \tag{7}$$

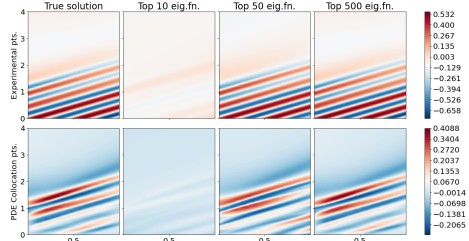

Notice that since the eigenfunctions depend on the entire eNTK matrix, including its off-diagonal terms, they naturally capture the cross information between each type of training point. Empirically, we find that the PINN eNTK eigenvalues falls quickly, allowing the dominant eigenfunctions (i.e., those with the largest eigenvalues) to effectively form a basis for the NN output in augmented space. This is consistent with results from past eNTK works for

Figure 1: Reconstruction of the PINN prediction values $F[\hat{u}_\theta](z)$, after 10k GD training steps, in $(x, \mathrm{s})$ and $(x, \mathrm{p})$ subspaces of the 1D Advection equation, using the dominant eNTK eigenfunctions (i.e., those with the largest eigenvalues).

vanilla NNs (Kopitkov & Indelman, 2020; Vakili et al., 2021). Figure 1 shows how the PINN prediction values $F[\hat{u}_\theta](z)$ in $(x, \mathrm{s})$ and $(x, \mathrm{p})$ subspaces can be reconstructed by just the top dominant eNTK eigenfunctions. Further discussion and more eNTK eigenfunction plots are in Appendix F.

The change in residual at a point $z \in \mathcal{Z}$, given training set $Z$ for a GD step with small enough $\eta$, is

$$\Delta R_{\theta_t}(z; Z) \triangleq R_{\theta_{t+1}}(z; Z) - R_{\theta_t}(z; Z) \approx -\eta \sum_{i=1}^{\infty} \lambda_{t,i} \underbrace{\langle \psi_{t,i}(Z), R_{\theta_t}(Z) \rangle_{\mathcal{H}_\Theta}}_{a_{t,i}(Z)} \psi_{t,i}(z) \tag{8}$$

Note that $\Delta R_{\theta_t}(z; Z)$ describes how the residual *at any point* $z$ (of any point type, and including points outside $Z$) is influenced by GD training on any training set $Z$ that can consist of *multiple* point types. Hence, $\Delta R_{\theta_t}(z; Z)$ describes the overall training dynamics for all point types.

---

[5]For compactness, we will overload the notations with $R_{\theta_t}(Z)$ and $\Theta_t(Z, Z')$ for cases of multiple inputs, and use $\Theta_t(Z)$ to refer to $\Theta_t(Z, Z)$.

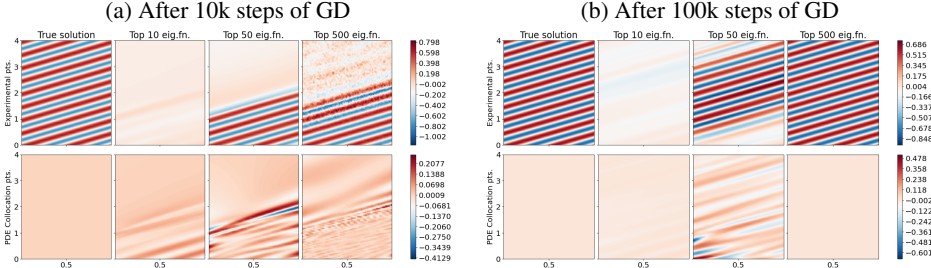

Figure 2: Reconstruction of the true PDE solution of the 1D Advection equation using eigenfunctions of $\Theta_t$ at different GD timesteps. For each time step, we plot the eigenfunction component both for $(x, \mathrm{s})$ subspace (top row) and $(x, \mathrm{p})$ subspace (bottom row).

We can gain useful insights on PINN training dynamics by considering the augmented eNTK eigenbasis. First, we show that the *residual components that align to the more dominant eigenfunctions will decay faster*. During a training period where the eNTK remains constant, i.e., $\Theta_t(z, z') \approx \Theta_0(z, z')$, the PINN residuals will evolve according to

$$R_{\theta_t}(z; Z) \approx R_{\theta_0}(z) - \Theta_0(z, Z) \sum_{i=1}^{\infty} \frac{1 - e^{-\eta t \lambda_{0,i}}}{\lambda_{0,i}} a_{0,i}(Z) \psi_{0,i}(Z) . \tag{9}$$

We provide a proof sketch of (9) in Appendix F. From (9), we can see that the residual component that is aligned to the eigenfunction $\psi_{t,i}$ (given by $a_{t,i}(Z)$) will decay as training time $t$ progresses at a rate proportional to $\eta \lambda_{0,i}$. For training set points $z \in Z$, the decay is exponential as (9) further reduces to $\sum_{i=1}^{\infty} e^{-\eta t \lambda_{0,i}} a_{t,i}(Z) \psi_{0,i}(Z)$. Interestingly, note that (9) applies to all $z \in \mathcal{Z}$, suggesting that choosing training point set $Z$ with residual components more aligned to dominant eigenfunctions also results in lower residuals for the entire augmented space and faster overall convergence.

Second, during PINN training, *the eNTK eigenfunctions will gradually align to the true target solution*. While many theoretical analysis of NTKs assumes a fixed kernel during training, this does not hold in practice (Kopitkov & Indelman, 2020). Figure 2 shows that the dominant eigenfunctions can better reconstruct the true PDE solution as training progresses, indicating that the eNTK evolves during training to further align to the target solution. Notice how the predictions on *both subspaces* needed to be considered for PINNs, which had not been explored before in past works. While the top 10 eigenfunctions replicated the $(x, \mathrm{p})$-subspace prediction better, more eigenfunctions were needed to reconstruct the full augmented space prediction (both $(x, \mathrm{s})$ and $(x, \mathrm{p})$ subspaces).

This suggests that PINN training consists of two concurrent processes of 1) *information propagation from the residuals* to the network parameters mainly along the direction of the dominant eNTK eigenfunctions, and 2) *eNTK eigenfunction evolution* to better represent the true PDE solution. In Sec. 5.2, we propose a point selection algorithm that accounts for *both* processes by selecting points that would cause the largest residual decrease to speed up Process 1, and re-calibrate the selected training points by re-computing the eNTK to accommodate Process 2.

### 4.3 CONVERGENCE DEGREE CRITERION AND RELATIONS TO GENERALIZATION BOUNDS

The change in residual $\Delta R_{\theta_t}(z; Z)$ (8) only captures the convergence at a single test point $z$. While it is possible to consider the average change in residual w.r.t. some validation set, the quality of this criterion would depend on the chosen set. Instead, to capture the convergence of the whole domain, we consider the function $\Delta R_{\theta_t}(\cdot; Z)$ in the reproducing kernel Hilbert Space (RKHS) $\mathcal{H}_{\Theta_t}$ of $\Theta_t$, with its inner product $\langle \cdot, \cdot \rangle_{\mathcal{H}_{\Theta_t}}$ defined such that $\langle \psi_{t,i}, \psi_{t,j} \rangle_{\mathcal{H}_{\Theta_t}} = \delta_{ij}/\lambda_{t,i}$. Ignoring factors of $\eta$, we define the *convergence degree* $\alpha(Z)$ to be the RKHS norm of $\Delta R_{\theta_t}(\cdot; Z)$, or

$$\alpha(Z) \triangleq \|\Delta R_{\theta_t}(\cdot; Z)\|_{\mathcal{H}_{\Theta_t}}^2 = \sum_{i=1}^{\infty} \lambda_{t,i}^{-1} \langle \Delta R_{\theta_t}(\cdot; Z), \psi_{t,i} \rangle_{\mathcal{H}_{\Theta_t}}^2 = \sum_{i=1}^{\infty} \lambda_{t,i} a_{t,i}(Z)^2. \tag{10}$$

Selecting training points that maximizes the convergence degree could speed up training convergence, which corresponds to Process 1 of the PINN training process. Hence, our point selection algorithm will be centered around choosing a training set $Z$ that maximizes $\alpha(Z)$.

**Relations to Generalization Bounds.**     We now theoretically motivate our criterion further by showing that training points that maximizes (10) will lead to lower generalization bounds.

**Theorem 1** (Generalization Bound, Informal Version of Theorem 2). *Consider a PDE of the form* (1). *Let* $\mathcal{Z} = [0,1]^d \times \{s,p\}$, *and* $S \subset \mathcal{Z}$ *be a i.i.d. sample of size* $N_S$ *from a distribution* $\mathcal{D}_S$. *Let* $\hat{u}_\theta$ *be a NN which is trained on* $S$ *by GD, with a small enough learning rate, until convergence. Then, there exists constants* $c_1, c_2, c_3 = \mathcal{O}\big(\text{poly}(1/N_S, \lambda_{\max}(\Theta_0(S))/\lambda_{\min}(\Theta_0(S)))\big)$ *such that with high probability over the random model initialization,*

$$\mathbb{E}_{x \sim \mathcal{X}}[|\hat{u}_{\theta_\infty}(x) - u(x)|] \leq c_1 \|R_{\theta_\infty}(S)\|_1 + c_2 \alpha(S)^{-1/2} + c_3. \tag{11}$$

Theorem 1 is proven in Appendix G. Our derivation extends the generalization bounds presented by Shu et al. (2022b), and takes into account the PINN architecture (which is captured by the eNTK) and the gradient-based training method, unlike past works (Mishra & Molinaro, 2021; De Ryck & Mishra, 2022). The term $\|R_{\theta_\infty}(S)\|_1$ is the training error at convergence of the training set, and can be arbitrarily small (Gao et al., 2023) which we also observe empirically. This leaves the generalization error to roughly be minimized when $\alpha(Z)$ is maximized. This suggests that training points that maximizes the convergence degree should also lead to lower generalization error.

## 5 PROPOSED METHOD

### 5.1 NYSTROM APPROXIMATION OF CONVERGENCE DEGREE

In this section, we propose a practical point selection algorithm based on the criterion presented in (10). During point selection, rather than considering the whole domain $\mathcal{Z}$, we sample a finite pool $Z_{\text{pool}} \subset \mathcal{Z}$ to act as candidate points for our selection process. In practice, we are unable to compute the eigenfunctions of the eNTK exactly. Hence, we estimate the eNTK eigenfunctions and corresponding eigenvalues $\psi_{t,i}$ and $\lambda_{t,i}$ using the Nystrom approximation (Williams & Seeger, 2000; Schölkopf & Smola, 2002). Specifically, given a reference set $Z_{\text{ref}}$ of size $p$, and the eigenvectors $\tilde{v}_{t,i}$ and eigenvalues $\tilde{\lambda}_{t,i}$ of $\Theta_t(Z_{\text{ref}})$, for each $i = 1, \ldots, p$, we can approximate $\psi_{t,i}$ and $\lambda_{t,i}$ by

$$\psi_{t,i} \approx \hat{\psi}_{t,i}(z) \triangleq \sqrt{|Z_{\text{ref}}|}\, \tilde{\lambda}_{t,i}^{-1}\, \Theta_t(z, Z_{\text{ref}})\tilde{v}_{t,i} \quad \text{and} \quad \lambda_{t,i} \approx \hat{\lambda}_{t,i} \triangleq |Z_{\text{ref}}|^{-1}\tilde{\lambda}_{t,i}. \tag{12}$$

In practice, selecting $p \ll |Z_{\text{pool}}|$ is sufficient since NTK eigenvalues typically decay rapidly (Lee et al., 2019; Wang et al., 2022c). Given $\hat{\psi}_{t,i}$ and $\hat{\lambda}_{t,i}$, we can approximate $a_{t,i}(Z)$ (8) as

$$a_{t,i}(Z) \approx \hat{a}_{t,i}(Z) \triangleq \langle \hat{\psi}_{t,i}(Z), R_{\theta_t}(Z) \rangle = \sqrt{|Z_{\text{ref}}|}\, \tilde{\lambda}_{t,i}^{-1}\, \tilde{v}_{t,i}^\top \Theta_t(Z_{\text{ref}}, Z) R_{\theta_t}(Z) \tag{13}$$

and subsequently approximate the convergence degree (10) as

$$\alpha(Z) \approx \hat{\alpha}(Z) \triangleq \sum_{i=1}^p \hat{\lambda}_{t,i}\, \hat{a}_{t,i}(Z)^2 = \sum_{i=1}^p \tilde{\lambda}_{t,i}^{-1}\big(\tilde{v}_{t,i}^\top \Theta_t(Z_{\text{ref}}, Z) R_{\theta_t}(Z)\big)^2. \tag{14}$$

In Proposition 1, we show that the approximate convergence degree (14) provides a sufficiently good estimate for the true convergence degree.

**Proposition 1** (Criterion Approximation Bound, Informal Version of Proposition 2). *Consider* $\alpha$ *and* $\hat{\alpha}$ *as defined in* (10) *and* (14) *respectively. Let* $N_{\text{pool}}$ *be the size of set* $Z_{\text{pool}}$. *Then, there exists a sufficiently large* $p_{\min}$ *such that if a set* $Z_{\text{ref}}$ *of size* $p \geq p_{\min}$ *is sampled uniformly randomly from* $Z_{\text{pool}}$, *then with high probability, for any* $Z \subseteq Z_{\text{pool}}$, *it is the case that*

$$|\alpha(Z) - \hat{\alpha}(Z)| \leq \mathcal{O}(N_{\text{pool}}/p) \cdot \|R_{\theta_t}(Z)\|_2^2. \tag{15}$$

Proposition 1 is formally stated and proven in Appendix H using Nystrom approximation bounds.

### 5.2 PINNACLE ALGORITHM

Given the theory-inspired criterion (14), we now introduce the PINNACLE algorithm (Algorithm 1). The algorithm consists of two alternating phases: the point selection phase, and the training phase. In the point selection phase, we generate a random pool of candidate points $Z_{\text{pool}}$ (Line 4), then aim to select a subset $Z \subset Z_{\text{pool}}$ which fits our constraints and maximizes $\hat{\alpha}(Z)$ (Line 6). However, this optimization problem is difficult to solve exactly. Instead, we propose two approaches to approximate the optimal set for $\hat{\alpha}$ (additional details are in Appendix I.1).

**SAMPLING (S).** We sample a batch of points from $Z_{\text{pool}}$ where the probability of selecting a point $z \in Z_{\text{pool}}$ is proportional to $\hat{\alpha}(z)$. This method will select points that individually have a high convergence degree, while promoting some diversity based on the randomization process.

**K-MEANS++ (K).** We represent each point with a vector embedding $z \mapsto \left( \hat{\lambda}_{t,i}^{1/2} \hat{a}_{t,i}(z) \right)_{i=1}^{p}$, and perform K-Means++ initialization on these embedded vectors to select a batch of points. Similar to previous batch-mode active learning algorithms (Ash et al., 2020), this method select points with high convergence degrees while also discouraging selection of points that are similar to each other.

---

**Algorithm 1** PINNACLE

1: **Input:** PINN $\hat{u}_\theta$, learning rate $\eta$, number of iterations $T$, eNTK appproximation error $\delta$.
2: **repeat**
3:     // Point selection phase
4:     Randomly sample candidates $Z_{\text{pool}}$ from $\mathcal{Z}$
5:     Compute $\Theta_t$ using Nystrom approximation
6:     Select subset $Z \subset Z_{\text{pool}}$ to fit constraint using SAMPLING or K-MEANS++
7:     // Training phase
8:     Compute $\bar{\Theta} = \Theta_t(Z_{\text{pool}})$
9:     **for** $t' = t, \ldots, t + T$ **do**
10:         $\theta_{t'+1} \leftarrow \theta_{t'} - \eta \nabla_\theta \mathcal{L}(\hat{u}_{\theta_{t'}}; Z)$
11:         Exit training if $\|\bar{\Theta} - \Theta_{t'}(Z_{\text{pool}})\| \geq \delta \|\bar{\Theta}\|$
12: **until** training converges or budget exhausted

---

In the training phase, we perform model training with GD (Line 10) for a fixed number of iterations or until the eNTK has change too much (Line 11). Inspired by Lee et al. (2019), we quantify this change based on the Frobenius norm of the difference of the eNTK since the start of the phase. This is to account for the eNTK evolution during GD (Process 2 in Sec. 4.2). The point selection phase will then repeat with a re-computed eNTK. In practice, rather than replacing the entire training set during the new point selection phase, we will randomly retain a portion of the old set and fill the remaining budget with new points selected based on the recomputed eNTK. We discuss these mechanisms further in Appendix I.2.

# 6 EXPERIMENTAL RESULTS AND DISCUSSION

## 6.1 EXPERIMENTAL SETUP

**Dataset.** For comparability with other works, we conducted experiments with open-sourced data (Takamoto et al., 2022; Lu et al., 2021), and experimental setups that matches past work (Raissi et al., 2019). The specific PDEs studied and details are in Appendix J.1.

**NN Architecture and Implementation.** Similar to other PINNs work, we used fully-connected NNs of varying NN depth and width with tanh activation, both with and without LAAF (Jagtap et al., 2020a). We adapted the DEEPXDE framework (Lu et al., 2021) for training PINNs and handling input data, by implementing it in JAX (Bradbury et al., 2018) for more efficient computation.

**Algorithms Benchmarked.** We benchmarked PINNACLE with a suite of non-adaptive and adaptive CL point selection methods. For our main results, apart from a baseline that randomly selects points, we compare PINNACLE with the HAMMERSLEY sequence sampler and RAD algorithm, the best performing non-adaptive and adaptive PDE CL point selection method based on Wu et al. (2023). We also extended RAD, named RAD-ALL, where IC/BC points are also adaptively sampled in the same way as the PDE CL points. We provide further details and results in Appendix J.3.

## 6.2 IMPACT OF CROSS INFORMATION ON POINT SELECTION

We first show how PINNACLE's automated point selection across all point types are surprisingly interpretable and similar to heuristics of past works that required manual tuning. To do so, we consider two time-dependent PDEs: (1) 1D Advection, which describes the motion of particles through a fluid with a given wave speed, and (2) 1D Burgers, a non-linear PDE for fluid dynamics combining effects of wave motion and diffusion. We experiment on the forward problem, whose goal is to estimate the PDE solution $u$ given the PDEs with known $\beta$ and IC/BCs but no EXP data.

Figure 3 plots the training points of all types selected by PINNACLE-K overlaid on the NN output $\hat{u}_\theta$, at various training stages. We can make two interesting observations. First, PINNACLE automatically adjusts the proportion of PDE, BC and IC points, starting with more IC CL points

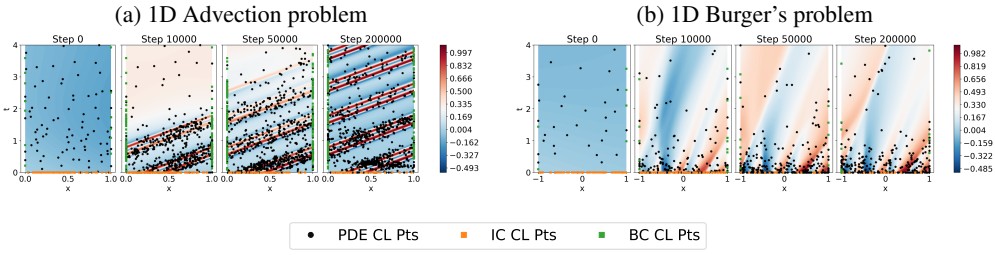

Figure 3: Example of points selected by PINNACLE-K during training of various forward problems. The points represent the various types of CL points selected by PINNACLE-K, whereas the patterns in the background represents the obtained NN solution.

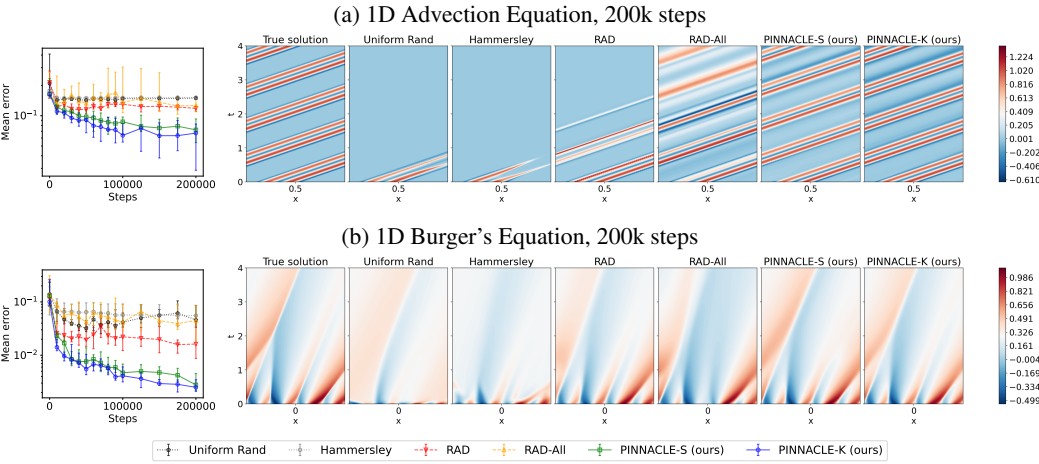

Figure 4: Results from the forward problem experiments. In each row, from left to right: plot of mean prediction error for each method, the true PDE solution, and the PINN output from different point selection methods.

and gradually shifting towards more PDE (and BC for Advection) CL points as training progresses. Intuitively, this helps as ICs provide more informative constraints for $\hat{u}_\theta$ in earlier training stages, while later on PDE points become more important for this information to "propagate" through the domain interior and satisfy the PDEs.

Second, the specific choice of points are surprisingly interpretable. For the Advection equation experiment (shown in Figure 3a), we can see that PINNACLE focuses on PDE and BC CL points closer to the initial boundary and "propagates" the concentration of points upwards, which helps the NN to learn the solution closer to the initial boundary before moving to later time steps. This may be seen as an automatic attempt to perform causal learning to construct the PDE solution (Wang et al., 2022b; Krishnapriyan et al., 2021), where earlier temporal regions are prioritized in training based on proposed heuristics. The selected points also cluster around more informative regions, such as the stripes in the Advection solution. Similar observations can be made for Burger's equation (shown in Figure 3b), where points focus around regions with richer features near the IC and sharper features higher in the plot. These behaviors translate to better training performance (Sec. 6.3).

## 6.3 EXPERIMENTAL RESULTS

**Forward Problems.** We present the results for the 1D Advection (Figure 4a) and 1D Burger's (Figure 4b) forward problem experiments. Quantitatively, both variants of PINNACLE are able to achieve a lower predictive loss than other algorithms. The performance gap is also large enough to be seen qualitatively, as PINNACLE variants learn significantly better solutions. This is especially

evident in 1D Advection (Figure 4a), where PINNACLE learn all stripes of the original solution while the benchmarks got stuck at only the first few closest to the ICs. Additional forward problem experiments and results can be found in Appendix K.2. For example, for a different problem setting where we provide more training points to other algorithms to achieve the same loss, we found that PINNACLE is able to reach the target loss faster. We also found that PINNACLE outperforms other point selection methods even in experiments where only one CL point type needs to be selected (e.g., hard-constrained PINNs), when no cross information among different point types is present.

**Inverse problem.** In the inverse problem, we estimate the unknown parameters of the PDE $\beta$, given the form of the PDE, IC/BC conditions, and some EXP points. We consider the 2D time-dependent incompressible Navier-Stokes equation, where we aim to find two unknown parameters in the PDE based on the queried EXP data. This is a more challenging problem consisting of coupled PDEs involving 3 outputs each with 2+1 (time) dimensions. Figure 5 shows that PINNACLE can recover the correct inverse parameters, while the other algorithms had difficulties doing so. Further results for the inverse problem can be found in Appendix K.3. For example, we demonstrate that its quicker convergence also allow PINNACLE to achieve a shorter computation time compared to other algorithms which are given a larger training budget to achieve comparable loss. We also showed how PINNACLE outperforms benchmarks for other complex settings, such as the inverse Eikonal equation problem where the unknown parameter is a 3D field rather than just scalar quantities.

**Transfer Learning of PINNs with Perturbed IC.** We also consider the setting where we already have a PINN trained for a given PDE and IC, but need to learn the solution for the same PDE with a perturbed IC. Different IC/BCs correspond to different PDE solutions, and hence can be viewed as different PINN tasks. To reduce cost, we consider fine-tuning the pre-trained PINN given a much tighter budget ($5\times$ fewer points) compared to training another PINN from scratch for the perturbed IC. Figure 6 shows results for the 1D Advection equation. Compared to other algorithms, PINNACLE is able to exploit the pre-trained solution structure and more efficiently reconstruct the solution for the new IC. Additional transfer learning results can be found in Appendix K.4.

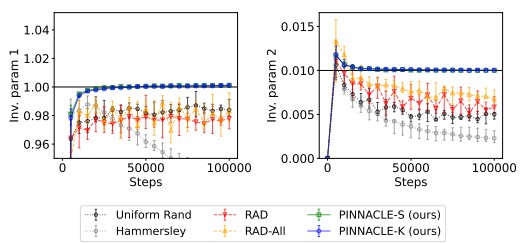

Figure 5: The two inverse parameters learned in the 2D Navier-Stokes equation problem. The black horziontal lines represents the true parameter values, while the other lines represent the predicted values of the inverse parameters during training.

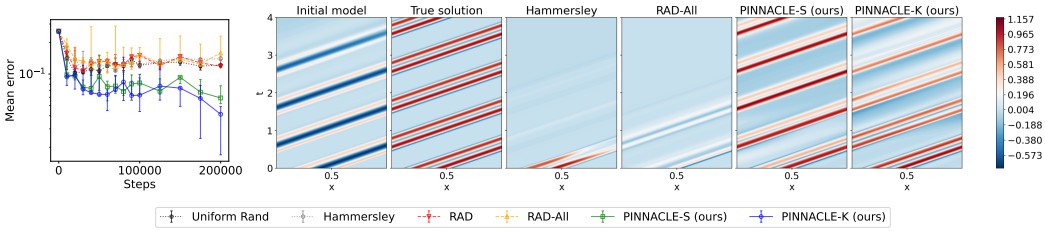

Figure 6: Predictive error for the 1D Advection transfer learning problem with perturbed IC. From left to right: plot of mean prediction error for each method, the true PDE solution, and the PINN output from different point selection methods.

## 7 CONCLUSION

We have proposed a novel algorithm, PINNACLE, that jointly optimizes the selection of all training point types for PINNs for more efficient training, while automatically adjusting the proportion of collocation point types as training progresses. While the problem formulation in this work is focused on PINNs, PINNACLE variants can also be applied to other deep learning problems which involve a composite loss function and input points from different domains as well. Future work could also involve extending PINNACLE to other settings such as deep operator learning.

## 8 REPRODUCIBILITY STATEMENT

The required assumptions and proof for Theorem 1 and Proposition 1 have been discussed in Appendix G and Appendix H respectively. The detailed experimental setup have been described in Appendix J. The code has been provided in `https://github.com/apivich-h/pinnacle`, with exception of PDEBENCH benchmarks which can be found on `https://github.com/pdebench/PDEBench`.

## ACKNOWLEDGEMENTS

This research/project is supported by the National Research Foundation, Singapore under its AI Singapore Programme (AISG Award No: AISG-PhD/2023-01-039J). This research is part of the programme DesCartes and is supported by the National Research Foundation, Prime Minister's Office, Singapore under its Campus for Research Excellence and Technological Enterprise (CREATE) programme.

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

## A    NOTATIONS AND ABBREVIATIONS

Table 1: List of notations used throughout the paper

| Symbol | Meaning | Example |
|---|---|---|
| $\delta_{i,j}$ | Dirac-Delta function, i.e., $\delta_{i,j} = 1$ if $i = j$ and 0 otherwise | |
| $\mathbb{1}$ | Identity matrix | |
| $\mathcal{N}$ | PDE operator | (1) |
| $\mathcal{B}$ | IC/BC operator | (1) |
| $\mathcal{X}$ | Input space of PDE solution | (1) |
| $\partial \mathcal{X}$ | Boundary of the input space of PDE solution | (1) |
| $u$ | PDE true solution | (1) |
| $\hat{u}_\theta$ | PINN used to approximate PDE solution | (2) |
| $\mathcal{L}(\hat{u}_\theta; X)$ | Loss of NN $\hat{u}_\theta$ w.r.t. set $X$ | (2) |
| $\eta$ | Learning rate | (3) |
| $x$ | Input of PDE solution, i.e., elements in $\mathcal{X}$ | |
| r | Generic training point label, i.e., $r \in \{s, p, b\}$ | |
| s, p, b | Specific training point label | (4) |
| $\mathcal{Z}$ | Augmented space | (4) |
| $Z$ | Set of elements from the augmented space | |
| $z$ | Point in augmented space | |
| $F$ | General prediction operator | (5) |
| $\Theta_t(\cdot, \cdot)$ | eNTK of NN $\hat{u}_{\theta_t}$ at timestep $t$ | (7) |
| $\lambda_{t,i}$ | $i$th largest eigenvalue of $\Theta_t$ | (7) |
| $\psi_{t,i}$ | Eigenfunction of $\Theta_t$ corresponding to eigenvalue $\lambda_{t,i}$ | (7) |
| $\nabla_\theta \hat{u}_{\theta_t}(x)$ | Jacobian of $\hat{u}_\theta(x)$ with respect to $\theta$ at $\theta = \theta_t$ | |
| $R_{\theta_t}(z)$ | Residual at time $t$ at point $z$ | (8) |
| $\Delta R_{\theta_t}(z; Z)$ | Change in residual at time $t$ at point $z$ when trained with $Z$ | (8) |
| $a_{t,i}(Z)$ | Residual expansion coefficient in dimension $i$ at time $t$ | (8) |
| $\mathcal{H}_\Theta$ | RKHS of the NTK | |
| $\langle \cdot, \cdot \rangle_{\mathcal{H}_\Theta}$ | Inner product of RKHS space | (8) |
| $\| \cdot \|_{\mathcal{H}_\Theta}$ | Norm of RKHS space, i.e., $\|f\|^2_{\mathcal{H}_\Theta} = \langle f, f \rangle_{\mathcal{H}_\Theta}$ | (10) |
| $Z_{\text{ref}}$ | Subset of $\mathcal{Z}$ for Nystrom approximation | (12) |
| $Z_{\text{pool}}$ | Subset of $\mathcal{Z}$ for point selection | Algorithm 1 |

Table 2: List of abbreviations used throughout the paper in alphabetical order

| Abbreviation | Meaning |
|---|---|
| BC | boundary condition |
| CL | collocation (as in collocation points) |
| DL | deep learning |
| EXP | experimental (as in experimental points) |
| eNTK | empirical neural tangent kernel |
| GD | gradient descent |
| IC | initial condition |
| NN | neural network |
| NTK | neural tangent kernel |
| PDE | partial differential equation |
| PINN | physics-informed neural network |
| RKHS | reproducing kernel Hilbert space |
| w.r.t. | with respect to |

## B    RELATED WORKS

Multiple works have examined the challenges of training PINNs, such as failure of PINNs to learn certain PDE solutions, particularly those with high frequency (Krishnapriyan et al., 2021; Rohrhofer et al., 2023). In response, there have been several proposals to adjust the training procedures of PINNs to mitigate this effect. This includes reweighing the loss function (Wang et al., 2022c), causal learning (Wang et al., 2022b) or curriculum learning approaches (Krishnapriyan et al., 2021). Most

relevant to this paper are works that looked into training points selection to improve PINNs training performance, though they have focused only on better selection of PDE CL points rather than joint optimization of the selection of EXP points and all types of PDE and IC/BC CL points. These works either adopt a non-adaptive approach (Wu et al., 2023), or an adaptive approach involving a scoring function based on some quantity related to the magnitude of the residual at each training point (Lu et al., 2021; Nabian et al., 2021; Wu et al., 2023; Peng et al., 2022; Zeng et al., 2022; Tang et al., 2023).

Outside of PINNs, the most similar problem to training point selection for PINNs would be the problem of deep active learning, which is a well-studied problem (Settles, 2012; Ren et al., 2021), with proposed algorithms based on diversity measures (Sener & Savarese, 2018; Ash et al., 2020; Wang & Ye, 2015) and uncertainty measures (Gal et al., 2017; Kirsch et al., 2019). Active learning can be viewed as an exploration-focused variant of Bayesian optimization (BO), which also utilizes principled uncertainty measures provided by surrogate models to optimize a black-box function (Garnett, 2022; Dai et al., 2023a; Shahriari et al., 2016).

A number of works have also incorporated NTKs to estimate the NN training dynamics to better select points for the NN (Wang et al., 2021b; 2022a; Mohamadi et al., 2022; Hemachandra et al., 2023). However, most of these deep active learning methods are unable to directly translate to scientific machine learning settings (Ren et al., 2023). Nonetheless, there have been limited works regarding active learning for PINNs, such as methods using the uncertainty computed from an ensemble of PINNs (Jiang et al., 2022).

In this work, we extensively used NTKs to analyze our neural network training dynamics. NTKs are also useful other tasks such as data valuation (Wu et al., 2022), neural architecture search (Shu et al., 2022a;b), multi-armed bandits (Dai et al., 2023b), Bayesian optimization (Dai et al., 2022), and uncertainty quantification (He et al., 2020). NTKs have also been used to study NN generalization loss (Cao & Gu, 2019) and also in relating NN training with gradient descent and kernel ridge regression (Arora et al., 2019a; Vakili et al., 2021).

## C  BACKGROUND ON PHYSICS-INFORMED NEURAL NETWORK

In this section we motivate and provide some intuition for the loss function (2) used in the training of PINNs for readers who may be unfamiliar with them.

In regular NN training, our goal is to train a model such that its output is as close to the true solution as possible. This can be quantified using mean squared error (MSE) loss between the true output $u(x)$ and NN output $\hat{u}_\theta(x)$ at some selected samples, which can simply be written as $\sum_x \left( \hat{u}_\theta(x) - u(x) \right)^2$, ignoring constant factors.

For PINNs, we also want to further constrain our solution such that they follow some PDE with IC/BC conditions such as (1). This can be seen as a constraint on the NN output, where we want the expected residual to be zero, i.e., considering only the PDE without the IC/BC, the constraint would be in the form $\mathcal{N}[\hat{u}_\theta](x) - f(x) = 0$. However, since enforcing a hard constraint, given by the PDE, on the NN is difficult to do, we instead incorporate a soft constraint that introduce a penalty depending on how far the NN diverges from a valid PDE solution. We would want this penalty (or regularization term in the loss function) to be of the form of $\mathbb{E}_x \left[ \left( \mathcal{N}[\hat{u}_\theta](x) - f(x) \right)^2 \right]$, where instead of performing the expectation exactly we sample some CL points to approximate the PDE loss term. This is a motivation behind using non-adaptive CL point selection method, which is seen as a way to approximate the expected residual, akin to the roles of quadrature points used in numerical integration.

Alternatively, some works have considered an adaptive sampling strategy where the CL points are selected where there is a higher residual. In this case, the expected residual plays a different role, which is to encourage the NN to learn regions of the solution with higher loss.

## D  BACKGROUND ON NEURAL TANGENT KERNELS

PINNACLE relies on analyzing NN training dynamics and convergence rates. To do so we require the use of Neural Tangent Kernels (NTKs), which is a tool which can be used to analyze general

NNs as well as PINNs. The empirical NTK (eNTK) of a NN $\hat{u}_{\theta_t}$ at step $t$ of the training process (Jacot et al., 2018),

$$\Theta_t(x, x') = \nabla_\theta \hat{u}_{\theta_t}(x) \, \nabla_\theta \hat{u}_{\theta_t}(x')^\top, \tag{16}$$

can be used to characterize its training behavior under gradient descent (GD). For a NN trained using GD with learning rate $\eta$ on training set $X$, the NN parameters $\theta_t$ evolves as $\theta_{t+1} = \theta_t - \eta \nabla_\theta \mathcal{L}(\hat{u}_{\theta_t}; X)$ while the model output evolves as $\hat{u}_{\theta_{t+1}}(x) = \hat{u}_{\theta_t}(x) - \eta \Theta_t(x, X) \nabla_{\hat{u}_\theta} \mathcal{L}(\hat{u}_{\theta_t}; X)$. Given an MSE loss $\mathcal{L}(\hat{u}_\theta; X) = \sum_{x \in X} (\hat{u}_\theta(x) - u(x))^2 / 2|X|$, the change in model output during training in this case can be written as

$$\Delta \hat{u}_{\theta_t}(x) = \hat{u}_{\theta_{t+1}}(x) - \hat{u}_{\theta_t}(x) = -\eta \Theta_t(x, X)\big(\hat{u}_{\theta_t}(X) - u(X)\big). \tag{17}$$

Past works (Arora et al., 2019b) had shown that in the infinite-width limit and GD with infinitesimal learning rate, the NN follows a linearized regime and the eNTK converges to a fixed kernel $\Theta$ that is invariant over training step $t$. Even though these assumptions may not fully hold in practice, many works (e.g., Arora et al. (2019b;a); Lee et al. (2019); Shu et al. (2022b)) have shown that the NTK still provides useful insights to the training dynamics for finite-width NNs.

## E    BACKGROUND ON REPRODUCING KERNEL HILBERT SPACE

In this section, we provide a brief overview required on the reproducing kernel Hilbert space (RKHS) for our paper. This section adapted from Schölkopf & Smola (2002).

Consider a symmetric positive semi-definite kernel $\mathcal{K} : \mathcal{X} \times \mathcal{X} \to \mathbb{R}$. We can define an integral operator $T_\mathcal{K}$ as

$$T_\mathcal{K}[f](\cdot) = \int_\mathcal{Z} f(x) \mathcal{K}(x, \cdot) d\mu(x). \tag{18}$$

We can compute the eigenfunctions and corresponding eigenvectors of $T_\mathcal{K}$, which are pairs of functions $\psi_i$ and non-negative values $\lambda_i$ such that $T_\mathcal{K}[\psi_i](\cdot) = \lambda_i \psi_i(\cdot)$. Typically, we also refer to the eigenvectors and eigenvalues of $T_\mathcal{K}$ as the eigenvectors and eigenvalues of $\mathcal{K}$. By Mercer's theorem, we can show that the kernel $\mathcal{K}$ can be written in terms of its eigenfunctions and eigenvalues as

$$\mathcal{K}(x, x') = \sum_{i=1}^\infty \lambda_i \psi_i(x) \psi_i(x'). \tag{19}$$

Based on this, we define the feature map as

$$\phi_i(\cdot) = \mathcal{K}(x, \cdot) = \sum_{i=1}^\infty \lambda_i \psi_i(x) \psi_i(\cdot)$$

and consequently the inner product such that $\langle \psi_i, \psi_j \rangle_{\mathcal{H}_\mathcal{K}} = \delta_{i,j} / \lambda_i$. This is so that we are able to write

$$\langle \phi(x), \phi(x') \rangle_{\mathcal{H}_\mathcal{K}} = \sum_{i=1}^\infty \sum_{j=1}^\infty \lambda_i \lambda_j \psi_i(x) \psi_j(x') \langle \psi_i(x), \psi_j(x') \rangle_{\mathcal{H}_\mathcal{K}} \tag{20}$$

$$= \sum_{i=1}^\infty \frac{\lambda_i^2 \psi_i(x) \psi_i(x')}{\lambda_i} = \sum_{i=1}^\infty \lambda_i \psi_i(x) \psi_i(x') = \mathcal{K}(x, x'). \tag{21}$$

From this, the RKHS $\mathcal{H}_\mathcal{K}$ of a kernel $\mathcal{K}$ would be given by a set of function with finite RKHS norm, i.e.,

$$\mathcal{H}_\mathcal{K} = \left\{ f \in L_2(\mathcal{X}) : \langle f, f \rangle_{\mathcal{H}_\mathcal{K}} = \sum_{i=1}^\infty \frac{\langle f, \phi_i \rangle_{\mathcal{H}_\mathcal{K}}^2}{\lambda_i} \leq \infty \right\}. \tag{22}$$

## F    EXTENDED DISCUSSION FOR SEC. 4.2

The contents in this section has been adapted from NTK related works (Lee et al., 2019) to match the context of NTK for PINNs.

**Calculations of (8).** From the gradient descent update formula with loss function given by (6), we can see that

$$\theta_{t+1} - \theta_t = -\eta \nabla_\theta \mathcal{L}(\hat{u}_{\theta_t}; Z) \tag{23}$$

$$= -\eta \nabla_\theta F[\hat{u}_{\theta_t}](Z)^\top \nabla_{F[\hat{u}_\theta]} \mathcal{L}(\hat{u}_{\theta_t}; Z) \qquad \text{by chain rule} \tag{24}$$

$$= -\eta \nabla_\theta F[\hat{u}_{\theta_t}](Z)^\top R_{\theta_t}(Z) \tag{25}$$

which, as $\eta \to 0$, can be rewritten with continuous time GD (where the difference between each time step is approximated as a derivative) as (Lee et al., 2019)

$$\partial_t \theta_t = -\eta \nabla_\theta F[\hat{u}_{\theta_t}](Z)^\top R_{\theta_t}(Z) \tag{26}$$

and so under continuous time GD, we can write the function output as

$$\Delta R_{\theta_t}(z; Z) \approx \partial_t R_{\theta_t}(z; Z) \qquad \text{as } \eta \to 0 \tag{27}$$

$$= \partial_t F[\hat{u}_{\theta_t}](z) \tag{28}$$

$$= \nabla_\theta F[\hat{u}_{\theta_t}](z) \, \partial_t \theta_t \qquad \text{by chain rule} \tag{29}$$

$$= -\eta \nabla_\theta F[\hat{u}_{\theta_t}](z) \, \nabla_\theta F[\hat{u}_{\theta_t}](Z)^\top \, R_{\theta_t}(Z) \tag{30}$$

$$= -\eta \Theta_t(z, Z) \, R_{\theta_t}(Z). \tag{31}$$

**Calculations of (9).** In this case, we consider the PINN under the linearized regime, where we approximate $F[\hat{u}_\theta]$ using the Taylor approximation around $\theta_0$,

$$F^{\text{lin}}[\hat{u}_\theta](z) = F[\hat{u}_{\theta_0}](z) + \nabla_\theta F[\hat{u}_{\theta_0}](z) \, (\theta - \theta_0), \tag{32}$$

and subsequently $R_\theta^{\text{lin}}(z) = F^{\text{lin}}[\hat{u}_\theta](z) - F[u](z)$. Under this regime, the eNTK remains constant at all $t$, i.e., $\Theta_t(z, z') = \Theta_0(z, z') = \nabla_\theta F[\hat{u}_{\theta_0}](z) \, \nabla_\theta F[\hat{u}_{\theta_0}](z')^\top$ (Lee et al., 2019), which also means that its eigenvectors and corresponding values will not evolve over time. It has been shown empirically both for normal NNs (Lee et al., 2019) and PINNs (Wang et al., 2022c) that this regime holds true for sufficiently wide neural networks.

For simplicity we let $\psi_i$ and $\lambda_i$ be the eigenfunctions and corresponding eigenvalues of $\Theta_0$ (where we drop the subscripts $t$ for simplicity). The change in residue of the training set can be estimated using continuous time GD $\Delta R_{\theta_t}^{\text{lin}}(Z; Z) \approx \partial_t R_{\theta_t}^{\text{lin}}(Z; Z)$, and based on (31) can be rewritten as

$$\partial_t R_{\theta_t}^{\text{lin}}(Z) = -\eta \Theta_0(Z) \, R_{\theta_t}^{\text{lin}}(Z) \tag{33}$$

which, when solved, gives

$$R_{\theta_t}^{\text{lin}}(Z) = e^{-\eta t \Theta_0(Z)} R_{\theta_0}^{\text{lin}}(Z). \tag{34}$$

To extrapolate this result to other values of $z \in \mathcal{Z}$, we can use the definition of $R_{\theta_t}^{\text{lin}}(Z)$ to see that

$$F^{\text{lin}}[\hat{u}_{\theta_t}](Z) - F[u](Z) = e^{-\eta t \Theta_0(Z)} \big( F^{\text{lin}}[\hat{u}_{\theta_0}](Z) - F[u](Z) \big) \tag{35}$$

and therefore

$$F^{\text{lin}}[\hat{u}_{\theta_t}](Z) = F[u](Z) + e^{-\eta t \Theta_0(Z)} \big( F^{\text{lin}}[\hat{u}_{\theta_0}](Z) - F[u](Z) \big) \tag{36}$$

$$= F^{\text{lin}}[\hat{u}_{\theta_0}](Z) - \big( \mathbb{1} - e^{-\eta t \Theta_0(Z)} \big) \big( F^{\text{lin}}[\hat{u}_{\theta_0}](Z) - F[u](Z) \big) \tag{37}$$

$$= F^{\text{lin}}[\hat{u}_{\theta_0}](Z) - \Theta_0(Z)\Theta_0(Z)^{-1} \big( \mathbb{1} - e^{-\eta t \Theta_0(Z)} \big) R_{\theta_0}^{\text{lin}}(Z) \tag{38}$$

$$= F^{\text{lin}}[\hat{u}_{\theta_0}](Z) + \nabla_\theta F[\hat{u}_{\theta_0}](Z) \underbrace{\nabla_\theta F[\hat{u}_{\theta_0}](Z)^\top \Theta_0(Z)^{-1} \big( e^{-\eta t \Theta_0(Z)} - \mathbb{1} \big) R_{\theta_0}^{\text{lin}}(Z)}_{\theta_t - \theta_0}$$

$$\tag{39}$$

which, through comparing the above with the form of the linearized PINN, gives us what $\theta_t - \theta_0$ is during training. Therefore, plugging this into (32),

$$F^{\text{lin}}[\hat{u}_{\theta_t}](z) = F^{\text{lin}}[\hat{u}_{\theta_0}](z) + \nabla_\theta F[\hat{u}_{\theta_0}](z)\nabla_\theta F[\hat{u}_{\theta_0}](Z)^\top \Theta_0(Z)^{-1} \big( e^{-\eta t \Theta_0(Z)} - \mathbb{1} \big) R_{\theta_0}^{\text{lin}}(Z)$$

$$\tag{40}$$

$$= F^{\text{lin}}[\hat{u}_{\theta_0}](z) + \Theta_0(z, Z)\Theta_0(Z)^{-1} \big( e^{-\eta t \Theta_0(Z)} - \mathbb{1} \big) R_{\theta_0}^{\text{lin}}(Z) \tag{41}$$

$$R_{\theta_t}^{\text{lin}}(z; Z) = R_{\theta_0}^{\text{lin}}(z; Z) + \Theta_0(z, Z)\Theta_0(Z)^{-1} \big( e^{-\eta t \Theta_0(Z)} - \mathbb{1} \big) R_{\theta_0}^{\text{lin}}(Z). \tag{42}$$

We can then decompose (42) into

$$R_{\theta_t}^{\text{lin}}(z; Z) = R_{\theta_0}^{\text{lin}}(z; Z) + \Theta_0(z, Z) \sum_{i=1}^{\infty} \frac{e^{-\eta t \lambda_i} - 1}{\lambda_i} \psi_i(Z) \psi_i(Z)^\top R_{\theta_0}^{\text{lin}}(Z). \tag{43}$$

**Eigenspectrum of the eNTK.** In this section, we elaborate further about the eigenspectrum of the eNTK. In Figure 7, we plot out the eigenfunctions $\psi_{t,i}$ of the eNTK for the 1D Advection forward problem at GD training steps $t = 10k$ and $t = 100k$. We can see that in the case of $t = 100k$ (when the PINN output $\hat{u}_\theta$ is already able to reconstruct the PDE solution well), the eigenfunctions are able to reconstruct the patterns that correspond to the true solution well. Note that due to the tight coupling between the true NN output and the PDE residual, the components of the eigenfunction for $(x, \text{s})$ and $(x, \text{p})$ are correlated and may share similar patterns. The dominant eigenfunctions will then tend to either have $\psi_{t,i}(x, \text{s})$ which aligns well with the true solution, or have $\psi_{t,i}(x, \text{p})$ which is flat (corresponding to the true PDE solution whose residual should be flat). In the less dominant eigenfunctions, the patterns will align worse with the true solution. Meanwhile, in the $t = 10k$ case, we see that the eigenfunctions will share some pattern with the true solution, but will not be as refined as for $t = 100k$. Furthermore, the less dominant eigenfunctions will be even noisier.

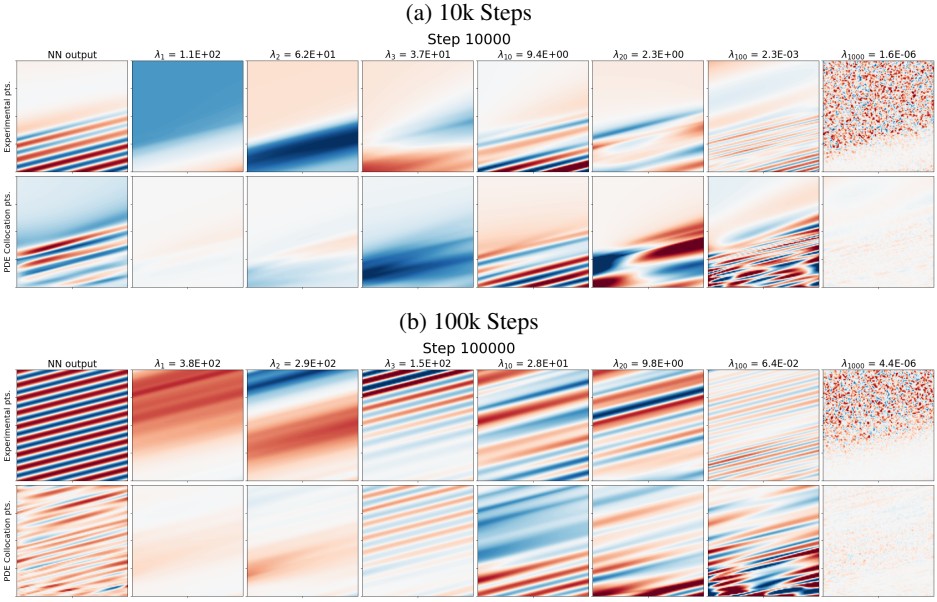

Figure 7: Eigenspectrum of $\Theta_t$ at various timesteps.

In Figure 8, we show how the eigenfunctions can describe the model training at a given GD training step. We see that by using just the top dominant eigenfunctions we are able to reconstruct the NN output $\hat{u}_\theta$.

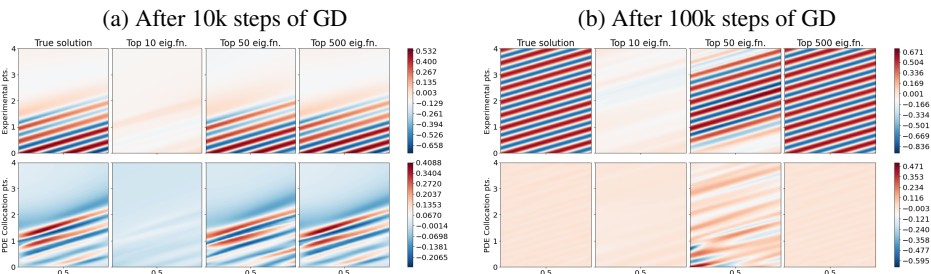

Figure 8: Reconstruction of the NN prediction using the eigenbasis of $\Theta_t$.

## G   PROOF ON GENERALIZATION BOUNDS

In this section, we aim to prove Theorem 1. The proof is an extension of Shu et al. (2022b) beyond regular NNs to give a generalization bound for PINNs. In order to do so, we will first state some assumptions about the PDE being solved and the PINN used for the training process.

**Assumption 1** (Stability of PDE). *Suppose we have a differential operator* $\mathcal{N} : \mathcal{P} \to \mathcal{Y}$ *where* $\mathcal{P}$ *is a closed subset of some Sobolev space and* $\mathcal{Y}$ *is a closed subset of some* $L^p$ *space. For any* $u, v \in \mathcal{P}$, *let there exists some* $C_{\mathrm{pde}} \geq 1$ *such that*

$$\|\mathcal{N}[u] - \mathcal{N}[v]\|_{\mathcal{Y}} \leq C_{\mathrm{pde}}\|u - v\|_{\mathcal{P}}. \tag{44}$$

For simplicity, we can view the Sobolev space as a $L^p$ space with a defined derivative up to a certain order. Note that while it is possible for $C_{\mathrm{pde}} < 1$, we assume otherwise for convenience.

We now provide an assumption about the NN we are training with.

**Assumption 2** (On the Neural Network Training). *Assume some NN* $\hat{u}_\theta : \mathcal{X} \to \mathbb{R}$ *with* $L$ *hidden layers each of width* $n_L$. *Suppose there exists constants* $\rho$ *such that* $\|\nabla_\theta \hat{u}_\theta(x)\| \leq \rho$ *and* $\|\nabla_\theta \mathcal{N}[\hat{u}_\theta](x)\| \leq \rho$. *Also let* $b_{\min}$ *and* $b_{\max}$ *be the minimum and maximum of the eigenvalues of the eNTK matrix of the possible training set at initialization, i.e., for any training set* $S$, $b_{\max} \geq \lambda_1(\Theta_0(S)) \geq \cdots \geq \lambda_{|S|}(\Theta_0(S)) \geq b_{\min}$.

We will also make an assumption about the boundedness of the PDE solution $u$ and the NN output $\hat{u}_\theta$. This is done for convenience of the proof – it is possible to add appropriate constants to the subsequent proofs to get a more general result.

**Assumption 3** (Boundedness of the NN Residual). *Suppose the PDE solution and the NN output was such that* $|R_\theta(x, \mathrm{s})| \leq 1/C_{\mathrm{pde}}$. *This would also imply that* $|R_\theta(x, \mathrm{p})| \leq 1$.

We will be working with the hypothesis spaces

$$\mathcal{U} \triangleq \{z \mapsto F[\hat{u}_{\theta_t}](z) : t > 0\} \tag{45}$$

and

$$\mathcal{U}^{\mathrm{lin}} \triangleq \{z \mapsto F[\hat{u}_{\theta_0}](z) + \nabla_\theta F[\hat{u}_{\theta_0}](z)\,(\theta_t - \theta_0) : t > 0\} \tag{46}$$

which is the linearization of the operated neural network $F[\hat{u}_\theta]$ around the initialization given by $F[\hat{u}_\theta](z) \approx F^{\mathrm{lin}}[\hat{u}_\theta](z) = F[\hat{u}_{\theta_0}](z) + \nabla_\theta F[\hat{u}_{\theta_0}](z)\,(\theta_t - \theta_0)$. Note that since $F^{\mathrm{lin}}$ performs a Taylor expansion with respect to $\theta$, this means the operator can only be applied on functions approximated by a NN that is parametrized by parameters $\theta$. This is unlike $F$ which is an operator that can be applied on any function. Also note that the linear expansion is performed on $F[\hat{u}_\theta]$ (rather than on just the operand $\hat{u}_\theta$), and hence does not require the assumption that $F$ is a linear operator. We can also define each components of $F^{\mathrm{lin}}[\hat{u}_\theta]$ separately as

$$\hat{u}_\theta^{\mathrm{lin}}(x) \triangleq F^{\mathrm{lin}}[\hat{u}_\theta](x, \mathrm{s}) = \hat{u}_{\theta_0}(x) + \nabla_\theta \hat{u}_{\theta_0}(x)\,(\theta_t - \theta_0), \tag{47}$$

and

$$\mathcal{N}^{\mathrm{lin}}[\hat{u}_\theta](x) \triangleq F^{\mathrm{lin}}[\hat{u}_\theta](x, \mathrm{p}) = \mathcal{N}[\hat{u}_{\theta_0}](x) + \nabla_\theta \mathcal{N}[\hat{u}_{\theta_0}](x)\,(\theta_t - \theta_0). \tag{48}$$

First we will state the result for training the NN where the loss function is based on the linearized operator, or

$$\mathcal{L}^{\mathrm{lin}}(\hat{u}_\theta; Z) = \frac{1}{2} \sum_{z \in Z} \left(F^{\mathrm{lin}}[\hat{u}_\theta](x) - F[u](x)\right)^2. \tag{49}$$

We will show that under the linearized regime, the weights of the NN will not change too much compared to its initialization.

**Lemma 1.** *Consider a linearized neural network* $\hat{u}_\theta^{\mathrm{lin}}$ *following Assumption 2 is trained on training set* $Z \subset \mathcal{Z}$ *with loss in the form* (49) *with GD with learning rate* $\eta < 1/\lambda_{\max}(\Theta_0(Z))$. *Then, for any* $t > 0$,

$$\|\theta_t - \theta_0\| \leq \|\theta_\infty - \theta_0\| = \sqrt{R_{\theta_0}(Z)^\top \Theta_0(Z)^{-1} R_{\theta_0}(Z)}. \tag{50}$$

*Proof.* Let $R_\theta^{\text{lin}}(z) = F^{\text{lin}}[\hat{u}_\theta](z) - F[u](z)$ be the residual of the linearized operated network. Note that $R_{\theta_0}^{\text{lin}}(z) = R_{\theta_0}(z)$. Based on the update rule of GD, we see that

$$\theta_{t+1} = \theta_t - \eta \cdot \nabla_\theta F[\hat{u}_{\theta_0}](Z)^\top R_{\theta_t}^{\text{lin}}(Z) \tag{51}$$

and so

$$F^{\text{lin}}[\hat{u}_{\theta_{t+1}}](Z) = F[\hat{u}_{\theta_0}](Z) + \nabla_\theta F[\hat{u}_{\theta_0}](Z)(\theta_{t+1} - \theta_0) \tag{52}$$

$$= F[\hat{u}_{\theta_0}](Z) + \nabla_\theta F[\hat{u}_{\theta_0}](Z)(\theta_t - \eta \cdot \nabla_\theta F[\hat{u}_{\theta_0}](Z)^\top R_{\theta_t}^{\text{lin}}(Z) - \theta_0) \tag{53}$$

$$= F^{\text{lin}}[\hat{u}_{\theta_t}](Z) - \eta \Theta_0 R_{\theta_t}^{\text{lin}}(Z) \tag{54}$$

$$= \left( \mathbb{1} - \eta \Theta_0 \right) F^{\text{lin}}[\hat{u}_{\theta_t}](Z) + \eta \Theta_0 F[u](Z) \tag{55}$$

$$= \left( \mathbb{1} - \eta \Theta_0 \right)^{t+1} R_{\theta_0}^{\text{lin}}(Z) + F[u](Z) \tag{56}$$

where $\mathbb{1}$ is the identity matrix and the shorthand $\Theta_0 = \Theta_0(Z)$. We note that (56) arrives from applying (55) recursively $t+1$ times and using the fact that we set $\eta < 1/\lambda_{\max}(\Theta_0(Z))$ to simplify a corresponding geometric series. This gives

$$\theta_t - \theta_0 = \sum_{k=0}^t (\theta_{k+1} - \theta_k) \tag{57}$$

$$= \eta \cdot \nabla_\theta F[\hat{u}_{\theta_0}](Z) \sum_{k=0}^t R_{\theta_k}^{\text{lin}}(Z) \tag{58}$$

$$= \eta \cdot \nabla_\theta F[\hat{u}_{\theta_0}](Z) \sum_{k=0}^t \left( \mathbb{1} - \eta \Theta_0 \right)^{t+1} R_{\theta_0}^{\text{lin}}(Z). \tag{59}$$

The remainder of the proof follows directly with Lemma A.5 from Shu et al. (2022b). □

**Corollary 1.** *Given Lemma 1,*

$$\|\theta_t - \theta_0\| \le \|R_{\theta_t}(S)\|^2 \cdot \sqrt{\frac{\lambda_{\max}(\Theta_0(S))}{\alpha(S) \cdot \lambda_{\min}(\Theta_0(S))}} \tag{60}$$

*Proof.* Notice that we can rewrite $\alpha(S)$ as

$$\alpha(S) = \left\langle \sum_{i=1}^\infty \lambda_{t,i} \langle \psi_{t,i}(S), R_{\theta_t}(S) \rangle \psi_{t,i}(S), \sum_{i=1}^\infty \lambda_{t,i} \langle \psi_{t,i}(S), R_{\theta_t}(S) \rangle \psi_{t,i}(S) \right\rangle_{\mathcal{H}_{\Theta_t}} \tag{61}$$

$$= \sum_{i=1}^\infty \lambda_{t,i} \langle \psi_{t,i}(S), R_{\theta_t}(S) \rangle^2 \tag{62}$$

$$= R_{\theta_t}(S)^\top \left( \sum_{i=1}^\infty \lambda_{t,i} \psi_{t,i}(S) \psi_{t,i}(S)^\top \right) R_{\theta_t}(S) \tag{63}$$

$$= R_{\theta_t}(S)^\top \Theta_t(S) R_{\theta_t}(S) \tag{64}$$

$$= R_{\theta_t}(S)^\top \Theta_0(S) R_{\theta_t}(S) \tag{65}$$

where we use $\Theta_t = \Theta_0$ due to the linearized NN assumption. Let $A = \Theta_0(S)$, $y = R_{\theta_t}(S)$ and $\kappa = \|R_{\theta_t}(S)\|$. Note that $A$ is a PSD matrix, and therefore we can define some $A^{1/2}$ such that $A = (A^{1/2})^\top (A^{1/2})$, and also some $A^{-1/2}$ such that $A^{-1} = (A^{-1/2})^\top (A^{-1/2})$.

$$(y^\top A y)(y^\top A^{-1} y) = \|A^{1/2} y\|_2^2 \cdot \|A^{-1/2} y\|_2^2 \tag{66}$$

$$\le \kappa^2 \|A^{1/2}\|_2^2 \cdot \kappa^2 \|A^{-1/2}\|_2^2 \tag{67}$$

$$= \kappa^4 \cdot \lambda_{\max}(A^{1/2})^2 \cdot \lambda_{\max}(A^{-1/2})^2 \tag{68}$$

$$= \kappa^4 \cdot \lambda_{\max}(A) \cdot \lambda_{\max}(A^{-1}) \tag{69}$$

$$= \kappa^4 \cdot \frac{\lambda_{\max}(A)}{\lambda_{\min}(A)} \tag{70}$$

which gives

$$\sqrt{R_{\theta_0}(S)^\top \Theta_0(S)^{-1} R_{\theta_0}(S)} \leq \|R_{\theta_t}(S)\|^2 \cdot \sqrt{\frac{\lambda_{\max}(\Theta_0(S))}{\alpha(S) \cdot \lambda_{\min}(\Theta_0(S))}}. \tag{71}$$

$\square$

We now will link the linearized operated NN to the non-linearized version. We first demonstrate that the difference of the two versions of the NN can be bounded.

**Lemma 2.** *Suppose we have a neural network $\hat{u}_\theta$ such that Assumption 2 holds. Then, there exists some constants $c, c', c'', N_L, \eta_0 > 0$ such that if $n_L > N_L$, then when applying GD with learning rate $\eta < \eta_0$,*

$$\sup_{t \geq 0} \|\hat{u}_{\theta_t}^{\mathrm{lin}} - \hat{u}_{\theta_t}\| \leq \frac{c}{\sqrt{n_L}} \tag{72}$$

*and*

$$\sup_{t \geq 0} \|\mathcal{N}^{\mathrm{lin}}[\hat{u}_{\theta_t}] - \mathcal{N}[\hat{u}_{\theta_t}]\| \leq \frac{c \cdot C_{\mathrm{pde}}}{\sqrt{n_L}} + c' \cdot \|R_{\theta_t}(S)\|^2 \cdot \sqrt{\frac{\lambda_{\max}(\Theta_0(S))}{\alpha(S) \cdot \lambda_{\min}(\Theta_0(S))}} \tag{73}$$

*with probability at least $1 - \delta$ over the random network initialization.*

*Proof.* The first part follows from Theorem H.1 of Lee et al. (2019). The second part follows the fact that

$$\|\mathcal{N}^{\mathrm{lin}}[\hat{u}_{\theta_t}](x) - \mathcal{N}[\hat{u}_{\theta_t}](x)]\| \tag{74}$$

$$= \|\mathcal{N}[\hat{u}_{\theta_0}](x) + \nabla_\theta \mathcal{N}[\hat{u}_{\theta_0}](x)^\top (\theta_t - \theta_0) - \mathcal{N}[\hat{u}_{\theta_t}](x)\| \tag{75}$$

$$\leq \|\mathcal{N}[\hat{u}_{\theta_0}](x) - \mathcal{N}[\hat{u}_{\theta_t}](x)\| + \|\nabla_\theta \mathcal{N}[\hat{u}_{\theta_0}](x)^\top (\theta_t - \theta_0)\| \tag{76}$$

$$\leq C_{\mathrm{pde}}\|\hat{u}_{\theta_0}(x) - \hat{u}_{\theta_t}(x)\| + \|\nabla_\theta \mathcal{N}[\hat{u}_{\theta_0}](x)^\top (\theta_t - \theta_0)\| \tag{77}$$

$$\leq C_{\mathrm{pde}}\left(\|\hat{u}_{\theta_0}(x) - \hat{u}_{\theta_t}^{\mathrm{lin}}(x)\| + \|\hat{u}_{\theta_t}^{\mathrm{lin}}(x) - \hat{u}_{\theta_t}(x)\|\right) + \|\nabla_\theta \mathcal{N}[\hat{u}_{\theta_0}](x)^\top (\theta_t - \theta_0)\| \tag{78}$$

$$\leq C_{\mathrm{pde}}\left(\|\nabla_\theta \hat{u}_{\theta_0}(x)^\top (\theta_t - \theta_0)\| + \|\hat{u}_{\theta_t}^{\mathrm{lin}}(x) - \hat{u}_{\theta_t}(x)\|\right) + \|\nabla_\theta \mathcal{N}[\hat{u}_{\theta_0}](x)^\top (\theta_t - \theta_0)\| \tag{79}$$

$$\leq C_{\mathrm{pde}}\|\nabla_\theta \mathcal{N}[\hat{u}_{\theta_0}](x)\,(\theta_t - \theta_0)\| + \|\nabla_\theta \mathcal{N}[\hat{u}_{\theta_0}](x)^\top (\theta_t - \theta_0)\| + C_{\mathrm{pde}}\|\hat{u}_{\theta_t}^{\mathrm{lin}}(x) - \hat{u}_{\theta_t}(x)\| \tag{80}$$

$$\leq \left(C_{\mathrm{pde}}\|\nabla_\theta \mathcal{N}[\hat{u}_{\theta_0}](x)\| + \|\nabla_\theta \mathcal{N}[\hat{u}_{\theta_0}](x)\|\right)\|\theta_t - \theta_0\| + C_{\mathrm{pde}} \cdot \frac{c}{\sqrt{n_L}} \tag{81}$$

$$\leq (1 + C_{\mathrm{pde}})\rho\|R_{\theta_t}(S)\|^2 \cdot \sqrt{\frac{\lambda_{\max}(\Theta_0(S))}{\alpha(S) \cdot \lambda_{\min}(\Theta_0(S))}} + C_{\mathrm{pde}} \cdot \frac{c}{\sqrt{n_L}} \tag{82}$$

which proves the second part of the claim. Note that the bound from (73) could be used to bound (72) as well. $\square$

We now use results above to compute the Rademacher complexity of $\mathcal{U}$. We first compute the Rademacher complexity of $\mathcal{U}^{\mathrm{lin}}$ in Lemma 3, then relate it to the Rademacher complexity of $\mathcal{U}$ in Lemma 4. As a reference, the empirical Rademacher of a hypothesis class $\mathcal{G}$ over a set $S$ is given by (Mohri et al., 2018)

$$\hat{\mathcal{R}}_S(\mathcal{G}) \triangleq \mathbb{E}_{\epsilon \in \{\pm 1\}^{|S|}} \left[ \sup_{g \in \mathcal{G}} \frac{\epsilon^\top g(S)}{|S|} \right]. \tag{83}$$

**Lemma 3.** *Let $\hat{\mathcal{R}}_S(\mathcal{G})$ be the Rademacher complexity of a hypothesis class $\mathcal{G}$ over some dataset $S$ with size $N_S$. Then, there exists some $c > 0$ such that with probability at least $1 - \delta$ over the random initialization,*

$$\hat{\mathcal{R}}_S(\mathcal{U}^{\mathrm{lin}}) \leq \frac{\rho}{N_S^{3/2}} \cdot \|R_{\theta_t}(S)\|^2 \sqrt{\frac{\lambda_{\max}(\Theta_0(S))}{\alpha(S) \cdot \lambda_{\min}(\Theta_0(S))}}. \tag{84}$$

*Proof.* We apply the same technique from Shu et al. (2022b). We see that

$$\hat{\mathcal{R}}_S(\mathcal{U}^{\text{lin}}) = \mathbb{E}_{\epsilon \sim \{\pm 1\}^{N_S}} \sum_{i=1}^{N_S} \frac{\epsilon_i F[\hat{u}_{\theta_0}](z_i)}{N_S} + \sup_{t \geq 0} \left[ \mathbb{E}_{\epsilon \sim \{\pm 1\}^{N_S}} \sum_{i=1}^{N_S} \frac{\epsilon_i \nabla_\theta F[\hat{u}_{\theta_0}](z_i)\, (\theta_t - \theta_0)}{N_S} \right] \tag{85}$$

$$\leq \sup_{t \geq 0} \frac{\|\theta_t - \theta_0\|_2 \|\nabla_\theta F[\hat{u}_{\theta_0}](S)\|_2}{N_S} \tag{86}$$

$$\leq \frac{1}{N_S} \cdot \|R_{\theta_t}(S)\|^2 \sqrt{\frac{\lambda_{\max}(\Theta_0(S))}{\alpha(S) \cdot \lambda_{\min}(\Theta_0(S))}} \cdot \frac{\rho}{\sqrt{N_S}}. \tag{87}$$

In the first equality, we use the definition of $\hat{\mathcal{R}}$. The first part of the first equality is independent of $t$, and will cancel to 0 due to the summation over $\epsilon$. The second inequality then follows from Cauchy-Schwartz inequality. The third inequality uses Lemma 1 and the assumption regarding $\|\nabla_\theta F[\hat{u}_{\theta_0}](z)\|$. $\qquad \square$

**Lemma 4.** *The Rademacher complexity of $\mathcal{U}$ is given by*

$$\hat{\mathcal{R}}_S(\mathcal{U}) \leq \frac{c \cdot C_{\text{pde}}}{\sqrt{n_L}} + \left( \frac{\rho}{N_S^{3/2}} + c' \right) \cdot \|R_{\theta_t}(S)\|^2 \sqrt{\frac{\lambda_{\max}(\Theta_0(S))}{\alpha(S) \cdot \lambda_{\min}(\Theta_0(S))}} \tag{88}$$

*for some $c$ and $c'$.*

*Proof.* From Lemma 2, we know that there will exist some constant $c$ and $c'$ such that for any $\epsilon_i \in \{\pm 1\}$,

$$\epsilon_i F[\hat{u}_{\theta_t}](z_i) \leq \epsilon_i F^{\text{lin}}[\hat{u}_{\theta_t}](z_i) + \frac{c \cdot C_{\text{pde}}}{\sqrt{n_L}} + c' \cdot \|R_{\theta_t}(S)\|^2 \cdot \sqrt{\frac{\lambda_{\max}(\Theta_0(S))}{\alpha(S) \cdot \lambda_{\min}(\Theta_0(S))}} \tag{89}$$

which gives

$$\hat{\mathcal{R}}_S(\mathcal{U}) \leq \hat{\mathcal{R}}_S(\mathcal{U}^{\text{lin}}) + \frac{c \cdot C_{\text{pde}}}{\sqrt{n_L}} + c' \cdot \|R_{\theta_t}(S)\|^2 \cdot \sqrt{\frac{\lambda_{\max}(\Theta_0(S))}{\alpha(S) \cdot \lambda_{\min}(\Theta_0(S))}}. \tag{90}$$

Applying Lemma 3 to the above completes the lemma. $\qquad \square$

We now use the Rademacher complexity to relate back to the overall residual. For simplicity, we let $r(z) = |R_{\theta_\infty}(z)|$ and let $\bar{r}_\mathcal{D} = \mathbb{E}_{z \sim \mathcal{D}}[r(z)]$ be the expected absolute residual given test samples drawn from a distribution $\mathcal{D}$.

**Lemma 5.** *Suppose $S \sim \mathcal{D}_S$ is an i.i.d. sample from $\mathcal{Z}$ according to some distribution $\mathcal{D}_S$ of size $N_S$. Then, there exists some $c, c', N_L > 0$ such that for $n_L \geq N_L$, with probability at least $1 - \delta$,*

$$\bar{r}_{\mathcal{D}_S} \leq \frac{1}{N_S} \|R_{\theta_\infty}(S)\|_1 + \frac{2c C_{\text{pde}}}{\sqrt{n_L}} + 2 \left( \frac{\rho}{N_S^{3/2}} + c' \right) \|R_{\theta_t}(S)\|^2 \sqrt{\frac{\lambda_{\max}(\Theta_0(S))}{\alpha(S) \cdot \lambda_{\min}(\Theta_0(S))}} + 3 \sqrt{\frac{\log(2/\delta)}{2N_S}} \tag{91}$$

*Proof.* Note that based on Assumption 3, we have $|R_{\theta_\infty}(z)| \leq 1$. The result then follows from directly applying Lemma 4 to Thm. 3.3 from Mohri et al. (2018). $\qquad \square$

We now proceed to prove Theorem 1. We will first state the theorem more formally, then provide a proof for it.

**Theorem 2** (Generalization Bound for PINNs). *Consider a PDE $\mathcal{N}[u, \beta](x) = f(x)$ which follows Assumption 1. Let $\mathcal{Z} = [0,1]^d \times \{\text{s}, \text{p}\}$. Let $\hat{u}_\theta$ be a NN which follows Assumption 2. Let the NN residual follows Assumption 3. Let $\mathcal{D}_S$ be a probability distribution over $\mathcal{Z}$ with p.d.f. $p(z)$ such that*

$\mu\big(\{z \in \mathcal{Z} : p(z) < 1/2\}\big) \le s$, *where $\mu(W)$ is the Lebesgue measure of set $W$[6]. Let $\hat{u}_\theta$ be trained on a dataset $S \subset \mathcal{Z}$, where $S \sim \mathcal{D}_S$ uniformly at random with $|S| = N_S$, by GD with learning rate $\eta \le 1/\lambda_{\max}(\Theta_0(S))$. Then, there exists some $c$ such that with probability $1 - 2\delta$ over the random model initialization and set $S$,*

$$\mathbb{E}_{x \sim \mathcal{X}}[|\hat{u}_{\theta_\infty}(x) - u(x)|] \le c_1 \|R_{\theta_\infty}(S)\|_1 + \frac{c_2}{\sqrt{\alpha(S)}} + c_3, \tag{92}$$

*where*

$$c_1 = \frac{2}{N_S}, \tag{93}$$

$$c_2 = 4\rho\sqrt{\frac{b_{\max}}{b_{\min}}}\left(\frac{1}{N_S^{3/2}} + (1 + C_{\text{pde}})\sqrt{\frac{b_{\max}}{b_{\min}}}\right), \tag{94}$$

$$c_3 = \frac{4c \cdot C_{\text{pde}}}{\sqrt{n_L}} + 6\sqrt{\frac{\log(2/\delta)}{2N_S}} + 2s. \tag{95}$$

*Proof.* Let $\mathcal{D}_{\mathcal{Z}}$ refer to the uniform distribution over $\mathcal{Z}$. The p.d.f. of the uniform distribution $q$ is given by $q(z) = 1/2$. Notice that we can then write

$$\bar{r}_{\mathcal{D}_{\mathcal{Z}}} = \frac{1}{2}\mathbb{E}_{x \sim \mathcal{X}}[r(x)] + \frac{1}{2}\mathbb{E}_{x \sim \mathcal{X}}[r(x)] \ge \frac{1}{2}\mathbb{E}_{x \sim \mathcal{X}}[r(x)]. \tag{96}$$

Let $\mathcal{Z}_\uparrow = \{z \in \mathcal{Z} : p(z) \ge 1/2\}$ be a set of input with a "high" probability of being chosen by the point selection algorithm, and $\mathcal{Z}_\downarrow = \mathcal{Z} \setminus \mathcal{Z}_\uparrow$. Then, abusing the integral notation to allow summation over discrete spaces, we see that

$$\bar{r}_{\mathcal{D}_{\mathcal{Z}}} = \mathbb{E}_{z \sim \mathcal{D}_{\mathcal{Z}}}[r(z)] \tag{97}$$

$$= \int_{\mathcal{Z}_\uparrow} r(z)dz + \int_{\mathcal{Z}_\downarrow} r(z)dz \tag{98}$$

$$\le \int_{\mathcal{Z}_\uparrow} r(z)dp(z) + \int_{\mathcal{Z}_\downarrow} r(z)dz \tag{99}$$

$$\le \int_{\mathcal{Z}} r(z)dp(z) + \int_{\mathcal{Z}_\downarrow} dz \tag{100}$$

$$= \bar{r}_{\mathcal{D}_S} + \mu(\mathcal{Z}_\downarrow) \tag{101}$$

$$\le \bar{r}_{\mathcal{D}_S} + s \tag{102}$$

and so

$$\mathbb{E}_{x \sim \mathcal{X}}[r(x)] \le 2(\bar{r}_{\mathcal{D}_S} + s). \tag{103}$$

Applying results from Lemma 5 to bound $\bar{r}_{\mathcal{D}_S}$ completes the proof. □

## H    FORMAL STATEMENT AND PROOF OF PROPOSITION 1

In this section, we will drop the subscript $t$ from the eNTK $\Theta_t$ for convenience.

---

[6]We use a slight abuse of notation to work with the augmented space $\mathcal{Z}$. Given $\mathcal{X}$ is defined as a Cartesian product of intervals, its Lebesgue measure is naturally defined. Then, formally, for $W \subseteq \mathcal{Z}$, we define $\mu(W) = \frac{1}{2}\mu(\{x \in \mathcal{X} : (x, \text{s}) \in W\}) + \frac{1}{2}\mu(\{x \in \mathcal{X} : (x, \text{p}) \in W\})$.

We first make the following observation about the quantity $\hat{\alpha}(Z)$ in (14). The quantity can be rewritten as

$$\hat{\alpha} = \sum_{i=1}^{p} \hat{\lambda}_{t,i}^{-1} \left( \hat{v}_i^\top \Theta(Z_{\text{ref}}, Z) R_{\theta_t}(Z) \right)^2 \tag{104}$$

$$= R_{\theta_t}(Z)^\top \Theta(Z, Z_{\text{ref}}) \left( \sum_{i=1}^{p} \frac{\hat{v}_i \hat{v}_i^\top}{\lambda_{t,i}} \right) \Theta(Z_{\text{ref}}, Z) R_{\theta_t}(Z) \tag{105}$$

$$= R_{\theta_t}(Z)^\top \underbrace{\Theta(Z, Z_{\text{ref}}) \Theta(Z_{\text{ref}})^{-1} \Theta(Z_{\text{ref}}, Z)}_{\text{③}} R_{\theta_t}(Z). \tag{106}$$

Notice that ③ can be seen as a Nystrom approximation of $\Theta(Z)$, where the approximation is done using the inputs $Z_{\text{ref}}$. We will first analyze ③, by comparing its difference from the full-rank matrix $\Theta(Z)$. This result is a direct consequence of the approximation error bounds from previous works on the Nystrom approximation method.

**Proposition 2.** *Let $N_{\text{pool}}$ be the size of set $Z_{\text{pool}}$. Let $\varepsilon \in (0,1)$ be a number. Suppose we write $\Theta(Z_{\text{pool}}) = U\Lambda U^\top$ using the eigenvalue decomposition such that columns of $U$ are eigenfunctions of $\Theta(Z_{\text{pool}})$ and diagonals of $\Lambda$ are their corresponding eigenvalues. Let $U_k$ be the first $k$ columns of $U$, corresponding to the top $k$ eigenvalues of $\Theta(Z_{\text{pool}})$, and let $\tau \triangleq (N_{\text{pool}}/k) \max_i \|(U_k)_i\|^2$ be the coherence of $U_k$. Let $k \leq N_{\text{pool}}$ be some integer. Then, if a set $Z_{\text{ref}}$ of size $p$ where*

$$p \geq \frac{2\tau k \log(k/\delta)}{(1-\varepsilon)^2} \tag{107}$$

*is sampled uniformly randomly from $Z_{\text{pool}}$, then with probability greater than $1 - \delta$, for any $Z \subseteq Z_{\text{pool}}$,*

$$\left\| \Theta(Z) - \Theta(Z, Z_{\text{ref}}) \Theta(Z_{\text{ref}})^{-1} \Theta(Z_{\text{ref}}, Z) \right\|_2 \leq \lambda_{k+1}(\Theta(Z_{\text{pool}})) \cdot \left( 1 + \frac{N_{\text{pool}}}{\varepsilon p} \right) \triangleq \hat{c} \tag{108}$$

*where $\lambda_k(\Theta(Z_{\text{pool}}))$ is the $k$th largest eigenvalue of $\Theta(Z_{\text{pool}})$.*

*Proof.* Let $D(Z) = \Theta(Z) - \Theta(Z, Z_{\text{ref}}) \Theta(Z_{\text{ref}})^{-1} \Theta(Z_{\text{ref}}, Z)$. Consider the case where $Z = Z_{\text{pool}}$. Then, based directly from Thm. 2 in Gittens (2011), we can show that $\|D(Z)\|_2 \leq \lambda_{\min}(\Theta(Z_{\text{pool}})) \cdot (1 + N_{\text{pool}}/\varepsilon p)$.

To apply this to every $Z$, note that since $Z \subseteq Z_{\text{pool}}$, it means $D(Z)$ will be a submatrix of $D(Z_{\text{pool}})$, and hence it is the case that $\|D(Z)\|_2 \leq \|D(Z_{\text{pool}})\|_2$, which completes the proof. □

A few remarks are in order.

*Remark* 1. The bound in Proposition 2 depends on the $k$th smallest eigenvalue of $\Theta(Z_{\text{pool}})$. From both empirical (Kopitkov & Indelman, 2020; Wang et al., 2022c) and theoretical (Vakili et al., 2021) studies of eNTKs, we find that the eigenvalue of eNTKs tend to decay quickly, meaning as long as our pool $Z_{\text{pool}}$ is large enough and we let $k$ be a large enough value, we would be able to achieve an approximation error which is arbitrarily small.

*Remark* 2. In Proposition 2 (and throughout the paper) we assume that the set $Z_{\text{ref}}$ is constructed by uniform random sample. As a result, the number of points $c$ that the bound requires is higher than what would be needed under other sampling schemes (Gittens & Mahoney, 2013; Wang et al., 2019), which we will not cover here. The version of Proposition 2 only serves to show that as long as we use enough points in $Z_{\text{ref}}$, the Nystrom approximation can achieve an arbitrarily small error in the eNTK approximation.

We now attempt to link the difference in Frobenius norm proven in Proposition 2 to the difference between $\alpha(Z)$ and $\hat{\alpha}(Z)$.

**Proposition 3.** *Let the conditions in Proposition 2. For a given $Z \subseteq Z_{\text{pool}}$, $|\alpha(Z) - \hat{\alpha}(Z)| \leq \hat{c}\|R_{\theta_t}(Z)\|_2^2$.*

*Proof.* It applies directly from Proposition 2 that

$$|\alpha(Z) - \alpha'(Z)| = \left| R_{\theta_t}(Z)^\top \big( \Theta(Z) - \Theta(Z, Z_{\text{ref}})\Theta(Z_{\text{ref}})^{-1}\Theta(Z_{\text{ref}}, Z) \big) R_{\theta_t}(Z) \right| \tag{109}$$

$$\leq \|R_{\theta_t}(Z)\|_2^2 \big\| \Theta(Z) - \Theta(Z, Z_{\text{ref}})\Theta(Z_{\text{ref}})^{-1}\Theta(Z_{\text{ref}}, Z) \big\|_2^2 \tag{110}$$

$$\leq \hat{c}\|R_{\theta_t}(Z)\|_2^2 \tag{111}$$

where the second inequality follows from the definition of Frobenius norm. $\qquad\square$

## I   EXTENDED DISCUSSION ON PINNACLE ALGORITHM

### I.1   DISCUSSION OF THE POINT SELECTION METHODS

**PINNACLE-S.** Theoretically, PINNACLE-S is better motivated than PINNACLE-K. This is because PINNACLE-S involves sampling from an implicit distribution (whose p.d.f. is proportional to $\hat{\alpha}$), and therefore is more directly compatible with assumptions required in Theorem 1, where $S$ is assumed to be i.i.d. samples from some unknown distribution.

In theory, PINNACLE-S is also sensitive to the ratio of the pool of points. For example, regions with many duplicate points will be more likely to be selected during sampling due to more of such points in the pool. This, however, is an inherent characteristic of many algorithms which selects a batch of points using sampling or greedy selection methods. Despite this, in our experiments we select a pool size that is around $4\times$ as large as the selected points budget in each round, which encourages enough diversity for the algorithm, while still allowing the algorithm to focus in on particular regions that may help with training. Future work could be done to make PINNACLE-S more robust to pool size, through using sampling techniques such as Gibbs sampling which would make the method independent of pool size.

**PINNACLE-K.** For the K-MEANS++ sampling mechanism, we use Euclidean distance for the embedded vectors. This is similar to the method used in BADGE Ash et al. (2020). Each dimension of the embedded vector will represent a certain eigenfunction direction (where a larger value represents a stronger alignment in this direction). Therefore, points that are selected in this method will tend to be pointing in different directions (i.e., act on different eigenfunction direction from one another), and also encourage selecting point that have high magnitude (i.e., high change in residual, far away from other points that has less effect on residual change).

We demonstrate this point further in Figure 9. In Figure 9a where we plot the vector embedding representations, we see that (i) the points with smaller convergence degree (i.e., lower norm in embedded space) tends to be "clustered" together close to the origin, meaning fewer of the points will end up being selected; (ii) the dimensions in the embedding space that correspond to the dominant eNTK eigenfunctions (i.e., higher eigenvalues) are more "stretched out", leading to coefficients that are more spread out, making points whose embedding align with these eigenfunctions to be further away from the other points. These two effects creates the "outlier" points that are more widely spread out due to having higher convergence degrees are preferentially selected. Figure 9b explicitly sorts the points based on their convergence degrees, and shows that PINNACLE-K mainly selects points that have much higher convergence degrees.

### I.2   TRAINING POINT RETENTION MECHANISM

We briefly describe how training points chosen in previous rounds can be handled during training. While it is possible to retain all of the training points ever selected, this can lead to high computational cost as the training set grows. Furthermore, past results (Wu et al., 2023) have also shown that retaining all of the training point could lead to worsen performances.

Alternatively, we can have a case where none of the training points are retained throughout training. This is done in the case of RAD algorithm (Wu et al., 2023). However, in practice, we find that this can lead to catastrophic forgetting, where the PINN forgets regions where training used to be at. As a result, in PINNACLE, we decide to retain a number of training points from the previous round when choosing the next training set. In Figures 10 and 11, we see that the performance of PINNACLE consistently drops when this mechanism is not used, demonstrating the usefulness of retaining past training points.

(a) Embedding of $\left(\hat{\lambda}_{t,i}^{1/2}\hat{a}_{t,i}(z)\right)_{i=1}^{p}$ for different values of $p$  (b) Value of $\hat{\alpha}(z)$

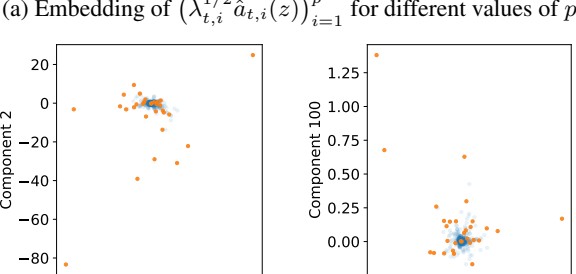 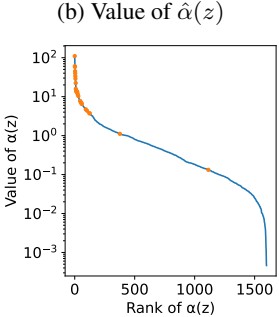

Figure 9: Illustrations of the K-Means++ sampling mechanism. In each plots, the blue points are the data pool, while the orange point represents the selected points by the K-Means++ mechanism. Figure 9a plots the embedding in two of the eigenfunction components, while Figure 9b plots the range of the individual convergence degree of each points. The embedding is obtained on the 1D Burger's equation problem after 100k trained steps, and the point selection process aims to select out 30 training points (note that this is fewer than the actual training set selected – we select fewer points to achieve a clearer illustration of the method).

### I.3 TRAINING TERMINATION VIA ENTK DISCREPANCY

In PINNACLE, we also check the eNTK and perform training point selection again when the eNTK changes too much. This corresponds to when the chosen set of training points are no longer optimal given the PINN training dynamics. In Figures 10 and 11, we show that auto-triggering of AL phase can help improve model training in some cases (in particular, for the Advection equation with PINNACLE-K as seen in Figure 11a). However, in practice, we also find that this is often less useful in practice, since as training progresses, the eNTK does not change by much and so the AL process will unlikely be triggered before the maximum training duration $T$ anyway.

### I.4 PSEUDO-OUTPUTS USED FOR EXP POINT SELECTION

The residual term $R_{\theta_t}(z)$ can be computed exactly in the case for CL points since the PDE and IC/BC conditions are completely known (except for the inverse problem where the PDE constants $\beta$ will need to be the estimated values at that point in training), and is independent of the true solution. However, in the case of EXP points, $R_{\theta_t}(x)$ will depend on the true output label, which is unknown during training. To solve this problem, we approximate the true solution $u$ by training the NN at the time of the point selection $\hat{u}_{\theta_t}$ for $T$ steps beyond that timestep, and use the trained network $\hat{u}_{\theta_{t+T}}$ as the pseudo-output. This works well in practice as the magnitude of the residual term roughly reflects how much error the model may have at that particular point based on its training dynamics. In our experiments, we set $T = 100$.

## J EXPERIMENTAL SETUP

### J.1 DESCRIPTION OF PDES USED IN THE EXPERIMENTS

In this section, we further describe the PDEs used in our benchmarks.

- **1D Advection equation.** The Advection PDE describes the movement of a substance through some fluid. This is a simple linear PDE which is often used to verify performance of PINNs in a simple and intuitive manner due to the easily interpretable solution. The 1D Advection PDE is given by

$$\frac{\partial u}{\partial t} - \beta\frac{\partial u}{\partial x} = 0, \quad x \in (0, L),\ t \in (0, T]$$  (112)

  with initial condition $u(x, 0) = u_0(x)$ and periodic BC $u(0, t) = u(L, t)$ is used. The closed form solution is given by $u(x, t) = u_0(x - \beta t)$. To compare with the PDE from (1),

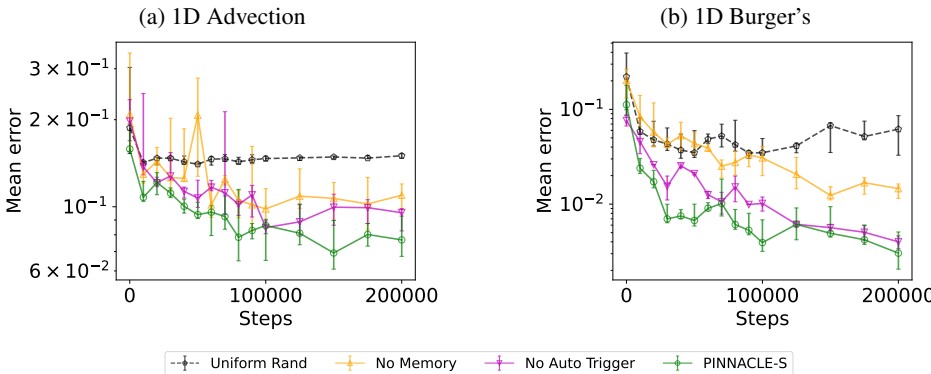

Figure 10: Performance of PINNACLE-S on the various forward problems when different components of the algorithm are removed.

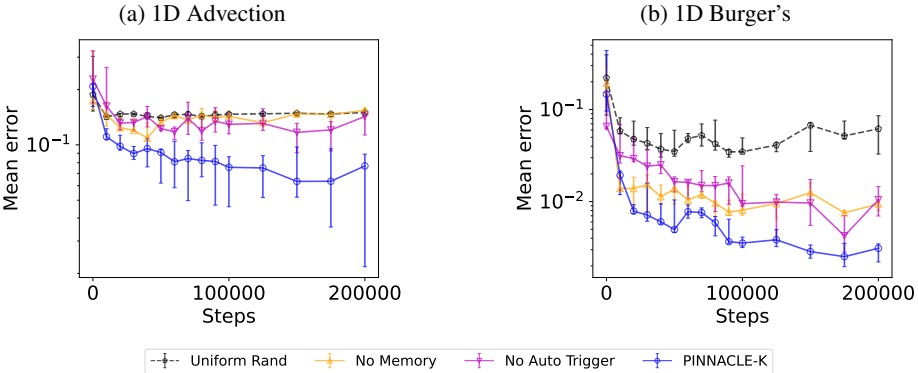

Figure 11: Performance of PINNACLE-K on the various forward problems when different components of the algorithm are removed.

we have $\mathcal{N}[u, \beta] = \frac{\partial u}{\partial t} - \beta \frac{\partial u}{\partial x}$ and $f(x, t) = 0$. Meanwhile, for the boundary conditions, the IC can be written as $\mathcal{B}_1[u] = u$ and $g_1(x, t) = u_0(x)$, while the BC can be written as $\mathcal{B}_2[u](x, t) = u(0, t) - u(L, t)$ and $g_2(x, t) = 0$.

In our forward problem experiments, we use $\beta = 1$ with a irregularly shaped IC from PDEBENCH (Takamoto et al., 2022). In the timing experiment (in Figure 21a), we use $\beta = 5$ with a sinusoidal initial condition, as past works have highlighted that the 1D Advection problem with high $\beta$ values is a challenging problem setting (Krishnapriyan et al., 2021), which is possible for the timing experiments given that we provide more CL points to the benchmark algorithms.

- **1D Burger's equation.** Burger's equation is an equation that captures the process of convection and diffusion of fluids. The solution sometimes can exhibit regions of discontinuity and can be difficult to model. The 1D Burger's equation is given by

$$\frac{\partial u}{\partial t} + \frac{1}{2} \frac{\partial u^2}{\partial x} = \nu \frac{\partial^2 u}{\partial x^2}, \quad x \in (0, L), \ t \in (0, T] \tag{113}$$

with initial condition $u(x, 0) = u_0(x)$ and periodic BC $u(0, t) = u(L, t)$ is applied. In our experiments, we used the dataset from Takamoto et al. (2022) with $\nu = 0.02$.

- **1D Diffusion Equation.** We aim to learn a neural network which solves the equation

$$\frac{\partial u}{\partial t} = \frac{\partial^2 u}{\partial x^2} - e^{-2t} \sin(4\pi x) \left(2 - 4\pi^2\right) \tag{114}$$

with initial conditions $u(x, 0) = \sin(4\pi x)$ and Dirichlet boundary conditions $u(-1, t) = y(1, t) = 0$. The reference solution in this case is $u(x, t) = e^{-2t} \sin(4\pi x)$.

In the experiments, we solve this problem by applying a hard constraint (Lagaris et al., 1998). To do so, we train a NN $\hat{u}'_\theta(x, t)$, however then transform the final solution according to $\hat{u}_\theta(x, t) = t(1 - x^2)\hat{u}'_\theta(x, t) + \sin(4\pi x)$.

- **1D Korteweg-de Vries (KdV) Equation.** We aim to learn the solution for the equation

$$\frac{\partial u}{\partial t} + \lambda_1 u\frac{\partial u}{\partial x} + \lambda_2 \frac{\partial^3 u}{\partial x^3} \tag{115}$$

with initial condition $u(x, 0) = \cos(\pi x)$ and periodic boundary condition $u(-1, t) = u(1, t)$. We solve the forward problem instance of the problem, where we set $\lambda_1 = 1$ and $\lambda_2 = 0.0025$.

- **2D Shallow Water Equation.** This is an approximation of the Navier-Stokes equation for modelling free-surface flow problems. The problem consists of a system of three PDEs modelling four outputs: the water depth $h$, the wave velocities $u$ and $v$, and the bathymetry $b$. In this problem, we are only interested in the accuracy of $h(x, y, t)$. We use the dataset as proposed by Takamoto et al. (2022).

- **2D Navier-Stokes Equation.** This is a PDE with important industry applications which is used to describe motion of a viscous fluid. In this paper, we consider the incompressible variant of the Navier-Stokes equation based on the inverse problem from Raissi et al. (2019), where we consider the fluid flow through a cylinder. The output exhibits a vortex pattern which is periodic in the steady state. The problem consists of the velocities $u(x, y, t)$ and $v(x, y, t)$ in the two spatial directions, and the pressure $p(x, y, t)$. The three functions are related through a system of PDEs given by

$$\frac{\partial u}{\partial t} + \lambda_1\left(u\frac{\partial u}{\partial x} + v\frac{\partial u}{\partial y}\right) = -\frac{\partial p}{\partial x} + \lambda_2\left(\frac{\partial^2 u}{\partial x^2} + \frac{\partial^2 u}{\partial y^2}\right), \tag{116}$$

$$\frac{\partial v}{\partial t} + \lambda_1\left(u\frac{\partial v}{\partial x} + v\frac{\partial v}{\partial y}\right) = -\frac{\partial p}{\partial y} + \lambda_2\left(\frac{\partial^2 v}{\partial x^2} + \frac{\partial^2 v}{\partial y^2}\right). \tag{117}$$

In the inverse problem experiments, we assume we are able to observe $u$ and $v$, and our goal is to compute the constants $\lambda_1$ and $\lambda_2$, as well as the pressure $p(x, y, u)$. There are no IC/BC conditions in this case.

- **3D Eikonal Equation.** The Eikonal equation is defined by

$$\|\nabla T(x)\| = \frac{1}{v(x)} \tag{118}$$

with $T(x_0) = 0$ for some $x_0$. For the inverse problem case, we are allowed to observe $T(x)$ at different values of $x$, and the goal is to reconstruct the function $v(x)$.

The budgets allocated for each types of problems are as listed below.

| Equation (Problem Type) | Training steps | CL points budget | Exp points query rate |
|---|---|---|---|
| 1D Advection (Fwd) | 200k | 1000 | - |
| 1D Burger's (Fwd) | 200k | 300 | - |
| 1D Diffusion (Fwd) | 30k | 100 | - |
| 1D KdV (Fwd) | 100k | 300 | - |
| 1D Shallow Water (Fwd) | 100k | 1000 | - |
| 2D Fluid Dynamics (Inv) | 100k | 1000 | 30 per 1k steps |
| 3D Eikonal (Inv) | 100k | 500 | 5 per 1k steps |
| 1D Advection (FT) | 200k | 200 | - |
| 1D Burger's (FT) | 200k | 200 | - |

## J.2 DETAILED EXPERIMENTAL SETUP

All code were implemented in JAX (Bradbury et al., 2018). This is done so due to its efficiency in performing auto-differentiation. To make DEEPXDE and PDEBENCH compatible with JAX, we made modifications to the experimental code. While DEEPXDE does have some support for JAX, it

remains incomplete for our experiments and we needed to add JAX code components for NN training and training point selection.

In each experiment, we use a multi-layer perceptron with the number of hidden layers and width dependent on the specific setting. Each experiment uses tanh activation, with some also using LAAF (Jagtap et al., 2020b). The models are trained with the Adam optimizer, with learning rate of $10^{-4}$ for Advection and Burger's equation problem settings, and $10^{-3}$ for the others. We list the NN architectures used in each of the problem setting below.

| Problem | Hidden layers | Hidden width | LAAF? |
|---------|:---:|:---:|:---:|
| 1D Advection (PDEBENCH, $\beta = 1$) | 8 | 128 | - |
| 1D Advection ($\beta = 5$) | 6 | 64 | Yes |
| 1D Burger's | 4 | 128 | - |
| 1D Diffusion (with hard constraints) | 2 | 32 | - |
| 1D Korteweg-de Vries | 4 | 32 | - |
| 2D Shallow Water | 4 | 32 | - |
| 2D Fluid Dynamics | 6 | 64 | Yes |
| 3D Eikonal | 8 | 32 | Yes |

The forward and inverse problem experiments in the main paper were repeated 10 rounds each, while the rest of the experiments are repeated 5 rounds each. In each graph shown in this paper, we report the median across these trials, and the error bars report the 20th and 80th percentile values.

For experiments where computational time is not measured, we performed model training on a compute cluster with varying GPU configurations. In the experiment where we measure the algorithms' runtime in forward problem (i.e., experiments ran corresponding to Figure 21a), we have trained each NN using one NVIDIA GeForce RTX 3080 GPU, and with Intel Xeon Gold 6326 CPU @ 2.90GHz, while for the inverse problem (i.e., experiments ran corresponding to Figures 21b and 21c), we trained each NN using one NVIDIA RTX A5000 GPU and with AMD EPYC 7543 32-Core Processor CPU.

### J.3 DESCRIPTION OF BENCHMARK ALGORITHMS

Below we list some of the algorithms that we ran experiments on. Note that due to space constraints, some of these experiments are not presented in the main paper, but only in the appendix.

We first list the non-adaptive point selection methods. All of these methods were designed to only deal with the selection of PDE CL points – hence IC/BC CL points and EXP points are all selected uniformly at random from their respective domains. The empirical performance of these methods have also been studied in Wu et al. (2023)

- UNIFORM RANDOM. Points are selected uniformly at random.
- HAMMERSLEY. Points are generated based on the Hammersley sequence.
- SOBOL. Points are generated based on the Sobol sequence.

The latter two non-adaptive point selection methods are said to be *low-discrepancy sequences*, which are often preferred over purely uniform random points since they will be able to better span the domain of interest. In the main paper we only report the results for HAMMERSLEY since it is the best pseudo-random point generation sequence as experimented by Wu et al. (2023).

We now list the adaptive point selection methods that we have used.

- RAD (Wu et al., 2023). The PDE CL points are sampled such that each point has probability of being sampled proportional to the PDE residual squared. The remaining CL points are sampled uniformly at random. This is the variant of the adaptive point selection algorithm reported by (Wu et al., 2023) which performs the best.
- RAD-ALL. This is the method that we extended from RAD, where the IC/BC CL points are also selected such that they are also sampled at probability proportional to the square of the residual. EXP points are selected using the same technique described in Appendix I.4,

which is comparable to expected model output change (EMOC) criterion used in active learning (Ranganathan et al., 2017).

- PINNACLE-S and PINNACLE-K. This is our proposed algorithm as described in Sec. 5.2.

In cases where there are multiple types of CL points, for all of the methods that do not use dynamic allocation of CL point budget, we run the experiments where we fix the PDE CL points at 0.5, 0.8 or 0.95 of the total CL points budget, and divide the remaining budget equally amongst the IC/BC CL points. In the main paper, the presented results are for when the ratio is fixed to 0.8.

# K    ADDITIONAL EXPERIMENTAL RESULTS

## K.1    POINT SELECTION DYNAMICS OF PINNACLE

In this section, we further consider the dynamics of the point selection process between different algorithms. Figure 12, we plot the point selection dynamics of various point selection algorithms in the 1D Advection equation problem. We can see that HAMMERSLEY will select points which best spans the overall space, however this is does not necessarily correspond to points which helps training the best. Meanwhile, RAD-ALL tends to focus its points on the non-smooth stripes of the solution, which corresponds to regions of higher residual. While this allows RAD-ALL to gradually build up the stripes, it does so in a less efficient manner than PINNACLE. In the variants of PIN-NACLE, we can see that the selected CL points gradually build upwards, which allows the stripe to be learnt more efficiently. We can also see that PINNACLE-K selects points that are more diverse than PINNACLE-S, which can be seen from the larger spread of CL points throughout training, with PDE CL points even attempting to avoid the initial boundary in Step 200k.

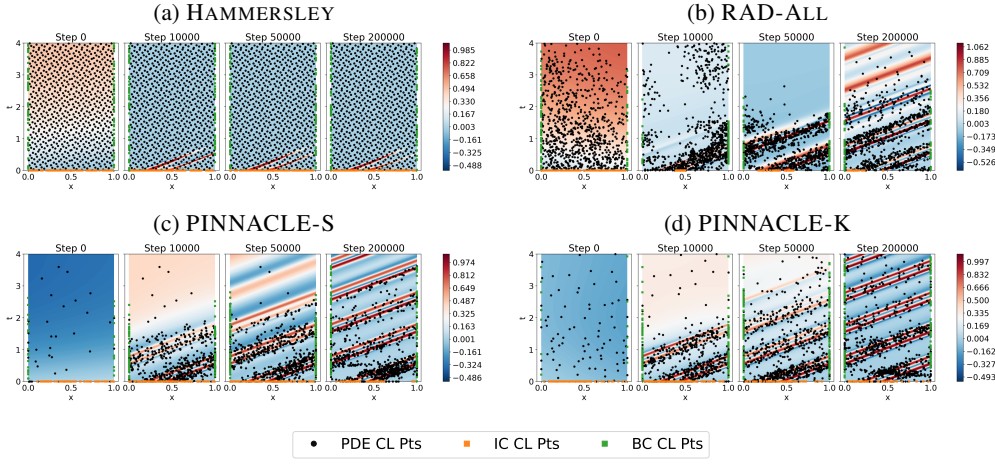

Figure 12: Training point selection dynamics for PINNACLE in the 1D Advection equation forward problem experiments.

In Figure 13, we plot the point selection dynamics of various point selection algorithms in the 1D Burger's equation problem. We can see that HAMMERSLEY peforms similar as in the Advection case, where it is able to span the space well but not able to guide model training well. Meanwhile, RAD-ALL selects points that are more spread out (that has high residual), but focuses less effort in regions near to the initial boundary or along the discontinuity in the solution, unlike PINNACLE. The points selection being less concentrated along these regions may explain why PINNACLE is able to achieve a better solution overall.

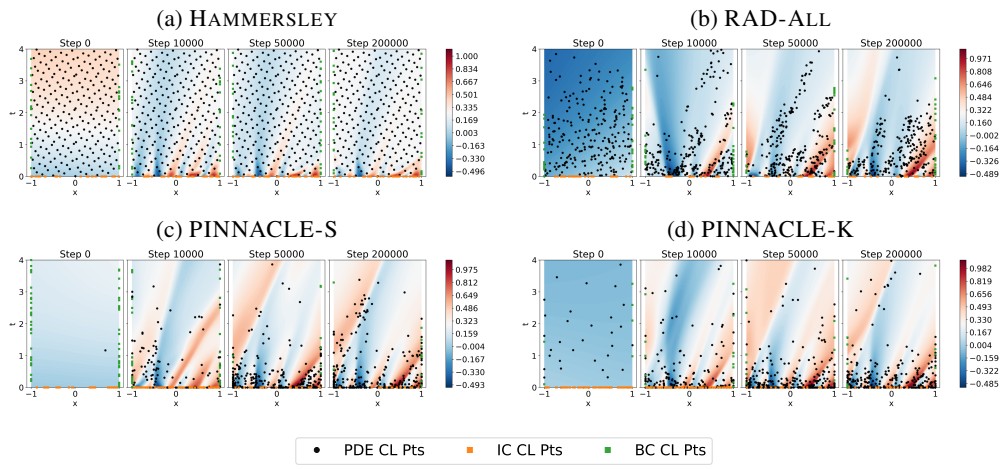

Figure 13: Training point selection dynamics for PINNACLE in the 1D Burger's equation forward problem experiments.

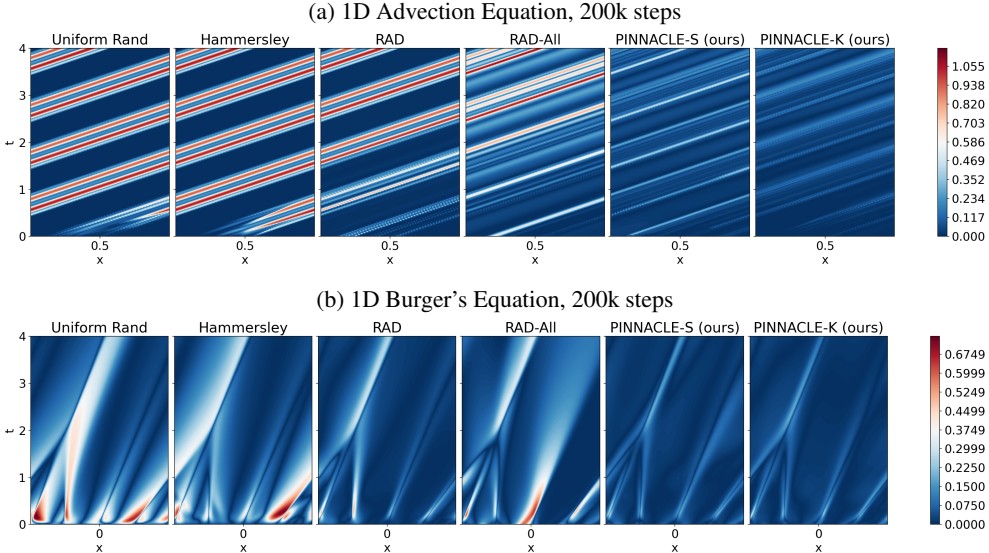

Figure 14: Predictive error for the forward problem experiments in Figure 4.

### K.2 FORWARD PROBLEMS

We plot the predictive error for experiments ran in Figure 4 in Figure 14. Additionally, we plot the 1D Advection and 1D Burger's equation experiments with additional point selection algorithms in Figures 15 and 16 respectively. Some observations can be made from these data. First, adaptive point selection methods outperform non-adaptive point selection methods. This is to be expected since adaptive methods allow for training points to be adjusted according to the current solution landscape, and respond better to the current training dynamics.

Second, for all the algorithms which do not perform dynamic CL point budget allocation, choosing whatever fixed budget at the beginning makes little difference in the training outcome. This can be seen that in each experiment, choosing the point allocation at 0.5, 0.8 or 0.95 shows little difference in performance, and is unable to outperform PINNACLE. Meanwhile, the variants of PINNACLE are able to exploit the dynamic budgeting to prioritize the correct types of CL points in different stages of training.

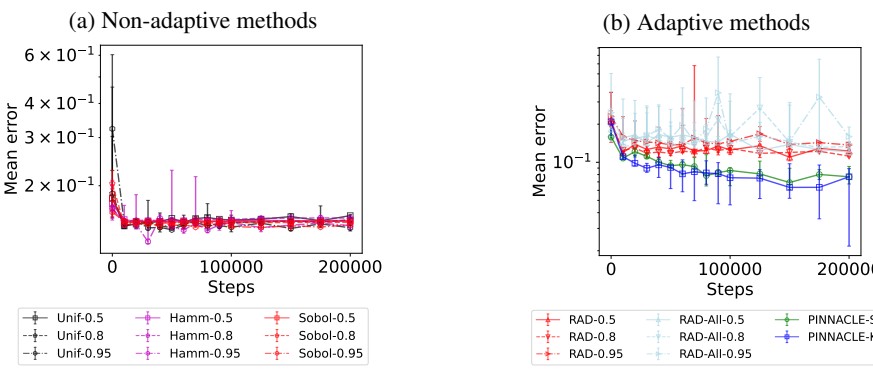

Figure 15: Further experimental results for 1D Advection equation forward problem.

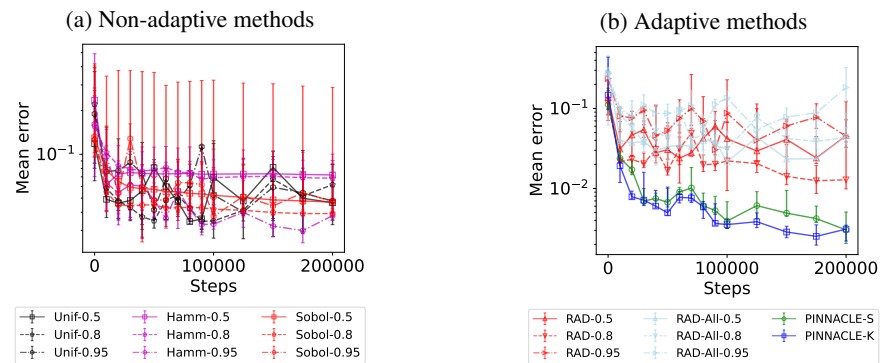

Figure 16: Further experimental results for 1D Burger's equation forward problem.

Third, while PINNACLE-K outperforms PINNACLE-S, it is often not by a large margin. This is an interesting result, since it may suggest that selecting a diverse set of points may not be required for training a PINN well.

Additionally, in Figure 21a, we also measure the runtime of different algorithms to train a 1D Advection problem with $\beta = 5$ until it reaches a mean loss of $10^{-2}$. We select this particular forward problem since it is known that Advection equation with a high $\beta$ is a difficult problem for PINN training (Krishnapriyan et al., 2021). We can see that while some algorithms are able to converge with higher number of training points, they will end up also incurring a larger runtime due to the higher number of training points. Meanwhile, PINNACLE-K is able to achieve a low test error with much fewer points than pseudo-random sampling at a much quicker pace. This shows that while it is possible to train PINNs with more CL points, this often come with computation cost of having to compute the gradient of more training data. PINNACLE provides a good way to reduce the number of training points needed for training, which reduces this training cost problem.

In Figure 18, we also ran the point selection for the forward 1D Diffusion equation where we apply a hard constraint to the PINN. This means the algorithm only requires selecting the PDE collocation point only. In the plots, we see that PINNACLE is able to still able to achieve better prediction loss than other methods, which shows that PINNACLE is still useful in collocation point selection even when the different training point types does not have to be considered.

## K.3 INVERSE PROBLEMS

In Figure 19, we present further experimental results for the Navier-Stokes equation experiments. We see that for the unseen components of the PDE solution (i.e., $p$, $\lambda_1$ and $\lambda_2$), PINNACLE is able to get much lower error values than the other algorithms (with exception of PINNACLE-K which is unable to recover $p$ as accurately). This shows that the point selection method is able to

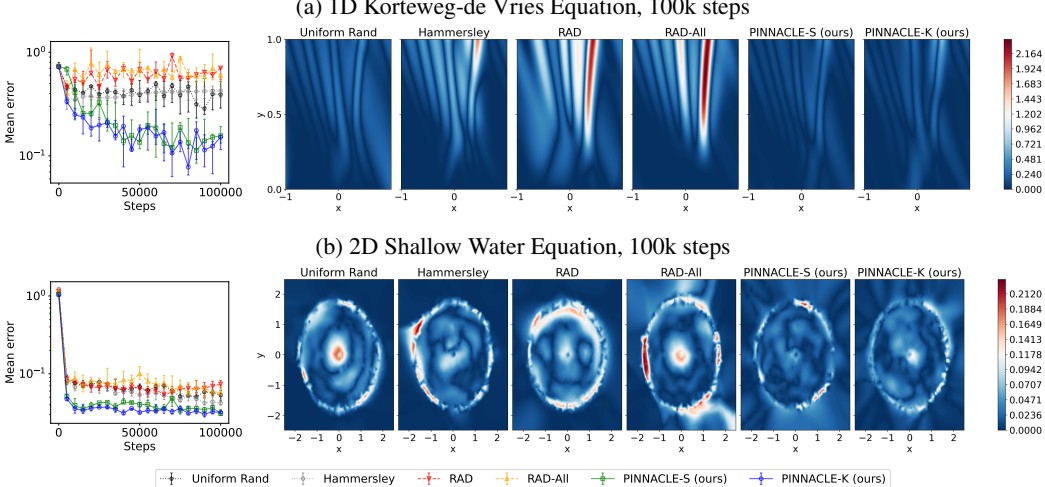

Figure 17: Results from additional forward problem experiments. In each row, we show the plot of mean prediction error for each method, and the PINN output error from different point selection methods. For Figure 17b, the right-hand plots show the error of $h(x, y, t)$ for $t = 1$.

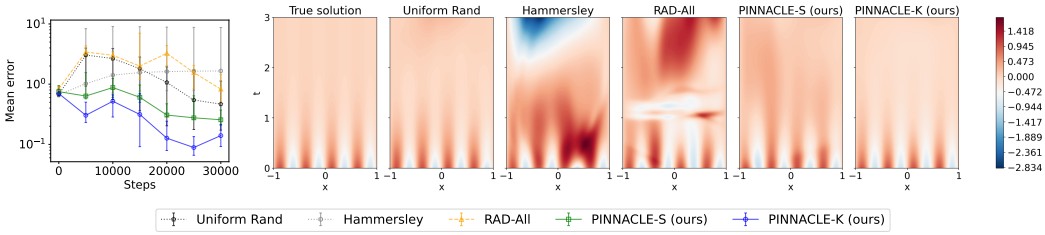

Figure 18: Results from the 1D Diffusion equation with hard-constrained PINN. The left-hand plot shows the mean prediction error for each method, while the right-hand plot shows the true value of the function and the predicted values from the PINN in the best trial.

select points to obtain the most information. For the prediction alone, all algorithms are able to perform about the same since they are able to get the information from the experimental points and interpolate accordingly. For the unknown quantities, however, a simple point selection mechanism fails whereas PINNACLE is able to perform better.

We also conduct additional inverse problem experiments on the Eikonal problem, as demonstrated in Figure 20. Unlike the Navier-Stokes problem whose aim is to learn the unknown constants of the PDE, the Eikonal inverse problem involves learning an unknown function $v(x, y, z)$ given the observed experimental readings $T(x, y, z)$. We see that PINNACLE is able to better recover $v(x, y, z)$ compared to the other point selection methods.

Furthermore, in Figure 21c, we measure the running time for the Navier-Stokes inverse problem where we aim to compute the PDE constants for the Navier-Stokes equation, and measure the time required until the estimated PDE constants are within 5% of the true value. We can see that using $10\times$ fewer points, PINNACLE-K outperforms the pseudo-random benchmark by being able to achieve the correct answer at a faster rate. Furthermore, since PINNACLE-K is able to converge with fewer steps, it means that in reality it would be much more efficient also since it requires less query for the actual solution, which saves experimental costs.

### K.4 TRANSFER LEARNING OF PINNs

In Figure 22, we compare the point selection dynamics for each point selection algorithms for the experiments from Figure 6. We can see that for PINNACLE, the CL points are selected in such a

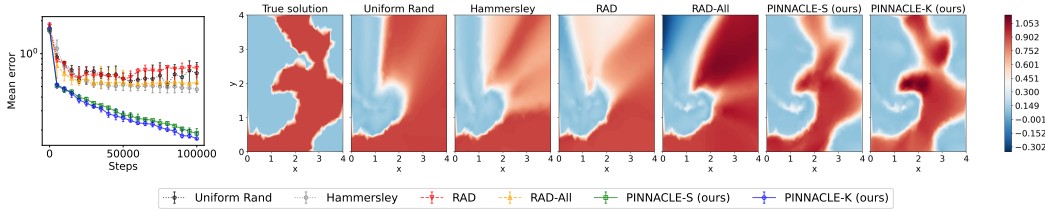

Figure 19: Errors of predicted value from the Navier-Stokes equation experiments. The graphs show, from left to right, the errors of overall prediction, the pressure $p$, $\lambda_1$, and $\lambda_2$.

Figure 20: Results from the 3D Eikonal equation inverse problem. The left-hand plot shows the mean prediction error for each method, while the right-hand plot shows the true value of $v(x, y, z)$ and the PINN output for learned $v(x, y, z)$ from different point selection methods when $z = 1$.

way that the existing solution structure still remains throughout training. This makes PINNACLE more efficient for transfer learning since it does not have to relearn the solution again from scratch. This is in contrast with the other methods, where the stripes eventually gets unlearned which makes the training inefficient.

In Figure 23, we present more results from the transfer learning experiments for the Burger's equation. As we can see, again, the variants of PINNACLE are able to better reconstruct the solution with the alternate IC than other methods.

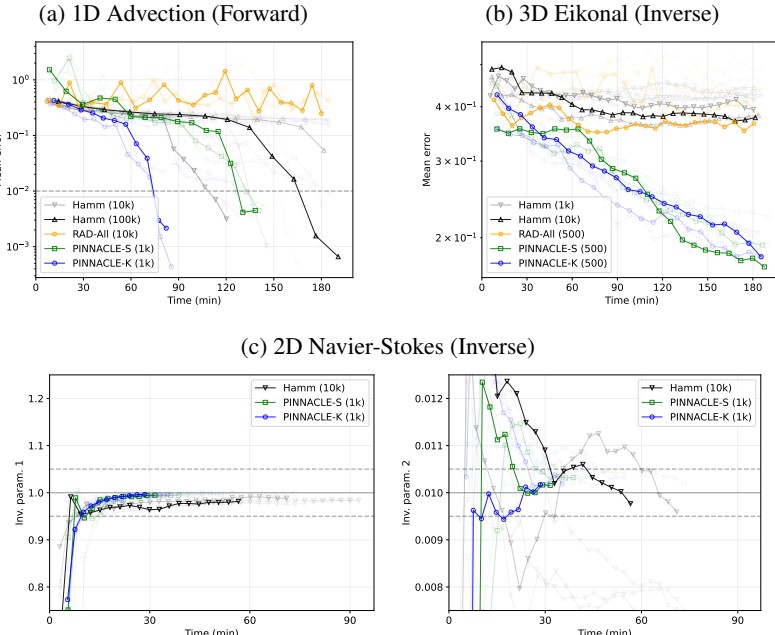

Figure 21: Experiments involving runtime of various point selection algorithms. The highlighted lines in each plot represent the example which is able to achieve the required accuracy and sustain the accuracy value for 5k additional steps within the shortest time. In Figures 21a and 21c, the dotted lines represent the threshold considered for acceptable accuracy – for Advection equation problem, this marks the $10^{-2}$ accuracy point, while for the Navier-Stokes inverse problem, this marks the region where the estimated constant are within 5% of the true value.

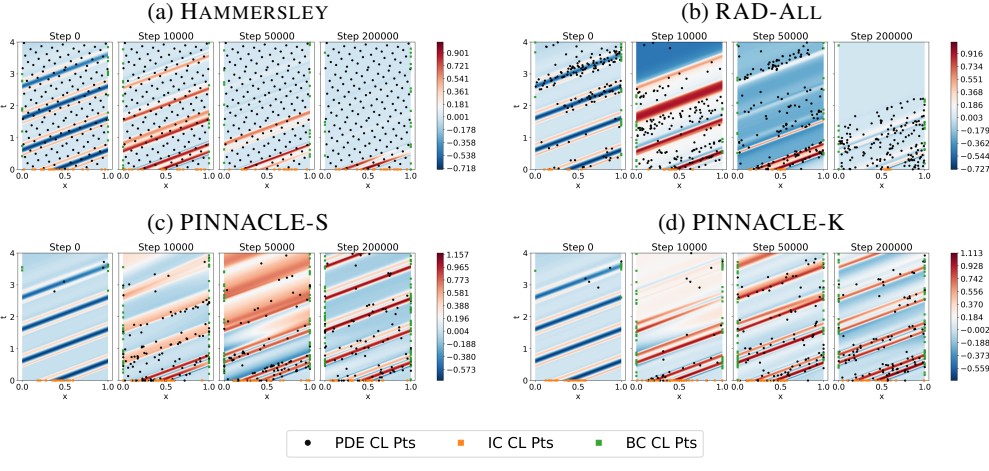

Figure 22: Training point selection dynamics for PINNACLE in the 1D Advection equation transfer learning problem experiments.

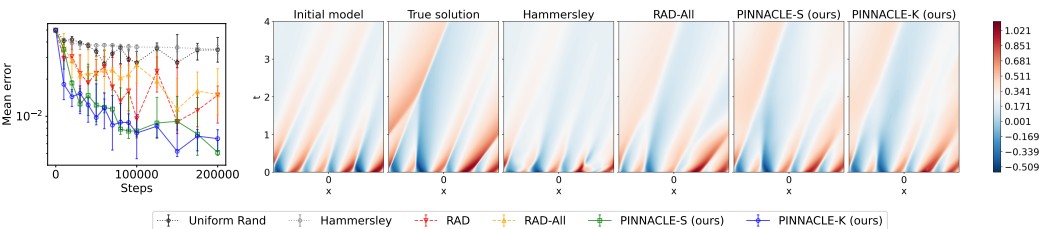

Figure 23: Predictive error for the 1D Burger's equation transfer learning problem with perturbed IC.

