# OpenReview forum: "PINNACLE: PINN Adaptive ColLocation and Experimental points selection"
_ICLR.cc/2024/Conference — ICLR 2024 spotlight_

### Official Review · Reviewer_nqGc · 2023-10-29

**Soundness:** 3 good
**Presentation:** 3 good
**Contribution:** 3 good
**Rating:** 8
**Confidence:** 4

**Summary:**

This work looks into adaptive experimental design for physics informed neural networks (PINNs) incorporating PDEs as constraints. The authors evaluate the design points from various types including experiments and initial and boundary conditions by computational constraint. They are then put together into the proposed PINN Adaptive Collocation and Experimental points selection, or PINNACLE, framework, which considers the interactions between the type of points selected. The authors theoretically demonstrate the relation of the framework to the generalization error, and the performance of PINNACLE compared to other design methods.

**Strengths:**

1. Experimental design for physics-informed neural networks (PINN) is a very interesting problem, and incorporating the PDE constraints and the initial and boundary conditions into the system makes the objective of the work quite attractive and carry substantial weight.
2. The notion of augmentation of points from various sources with the computational budget constraint is a unique idea that combines domain knowledge with experimental design strategies, which I think is of value.
3. The work expressed the contribution and introduced the proposed framework clearly with illustrations that are quite helpful for understanding.

**Weaknesses:**

1. Not necessarily a weakness, but I am quite interested in the selection procedure of $Z_{pool}$ from $Z$ as introduced in Algorithm 1 and whether the authors have considered some structured designs that may give it more advantage than random sampling.
2. Similar question regarding the selection procedure from subset $Z$ of $Z_{pool}$. I see two different design approaches are used here, but I wonder whether the authors have considered additional options, say weighted space-filling designs. Just a thought.

**Questions:**

1. My understanding is that the constraints by initial conditions or boundary conditions are only considered as "soft" constraints here due to how the objective function is set up. I guess for most situations its impact on the results would not be significant, or in other words, the approximation is going to be quite good. However, I wonder whether there could be cases where even a minor violation of the limiting conditions will cause significant distruptions to the system, and if so, how the authors propose to tackle that.
2. This follows my previous question. I see the limiting conditions are introduced as regularization components into the objective function, with the corresponding regularizer coefficients described as having a certain value. I wonder whether the authors have considered how to best place values on these regularizers according to the physical system, and whether that would have a substantial influence on the final result. I presume different limiting conditions will have varying impact on the system, so I am curious on whether the selection of coefficients may be incorporated into the framework.

---

> ### Author Response · Authors · 2023-11-17
> **Response to Reviewer nqGc (Part 1/2)**
>
> We would like to thank the reviewer for taking the time to review our work, and greatly appreciate the positive and insightful comments for our paper. We have provide clarifications to your comments and questions below.
>
> ---
>
> > I wonder whether there could be cases where even a minor violation of the limiting conditions will cause significant disruptions to the system, and if so, how the authors propose to tackle that.
>
> We would like to highlight the **transfer learning setting** that we presented (Sec. 6.3 in the main paper, Fig. 6), which **empirically demonstrates how PINNACLE may be also able to take into account the situation that you have described and still optimize its point selection**.
>
> - In this setting, we consider having access to a pre-trained PINN that had been trained for a PDE with a given initial condition (IC), but instead want a solution for the same PDE but with a perturbed IC. In the case of the Advection problem, as seen in Fig. 6, **a small perturbation in the IC results in relatively large global changes in the true solution** which can be qualitatively observed in the plots.
>
> - From Fig. 6, we can see that PINNACLE is able to **use a small number of points to fine-tune the pre-trained PINN to the perturbed IC conditions, and outperform other benchmarks.**
>
> -  In particular, we would like to draw your attention to Fig. 18 in Appendix K.4, which plots out the point selections of the two PINNACLE variants as well as that of the benchmarks. Notice how **both PINNACLE-S and PINNACLE-K were able to focus on the perturbed IC locations early on in training**, and as a result were able to more quickly learn the new solution with just very limited number of points.
>
> Hence, the results above provide some preliminary evidence that **PINNACLE is able to identify situations where minor changes in the IC/BC limiting conditions can have large effects on the final solution, and select training points** accordingly to learn the right solutions. We thank the reviewer for the insightful question, which could be investigated further in future works.
>
> ---
>
> > This follows my previous question. I see the limiting conditions are introduced as regularization components into the objective function, with the corresponding regularizer coefficients described as having a certain value. I wonder whether the authors have considered how to best place values on these regularizers according to the physical system, and whether that would have a substantial influence on the final result.
> > I presume different limiting conditions will have varying impact on the system, so I am curious on whether the selection of coefficients may be incorporated into the framework.
>
> While we did not focus on varying loss term coefficients for the different training point types in this work to simplify the presentation, we would like to highlight that our PINNACLE algorithm could readily do so and **incorporate any prior domain knowledge on the relative importance of the loss terms without any change to the algorithm design**. The different loss term coefficients set by the domain experts would be directly incorporated in the computation of the empirical Neural Tangent Kernel (eNTK) and effectively reweigh the residual terms, which will then impact the convergence degree computation and subsequently the choice of points. Training point types with larger loss term coefficients will therefore then likely be prioritized during the point selection process.
>
> ---
>
> &#8595; **Continued below** &#8595;

---

> ### Author Response · Authors · 2023-11-17
> **Response to Reviewer nqGc (Part 2/2)**
>
> > Not necessarily a weakness, but I am quite interested in the selection procedure of $Z_{pool}$ from $Z$ as introduced in Algorithm 1 and whether the authors have considered some structured designs that may give it more advantage than random sampling.
>
> > Similar question regarding the selection procedure from subset $Z$ of $Z_{pool}$. I see two different design approaches are used here, but I wonder whether the authors have considered additional options, say weighted space-filling designs. Just a thought.
>
> We thank the reviewer for the interesting suggestions. Indeed, one possibility is to incorporate low-discrepancy sequence sampling (e.g. Hammersley sequence sampler) to potentially construct a more diverse candidate pool $Z_{pool}$, which may lead to better point diversity for both the PINNACLE-S and PINNACLE-K variants, or serve as a foundation for other convergence degree-based heuristics.
>
> In this work, while we have proposed two variants of PINNACLE based on different point selection heuristics (SAMPLING and K-MEANS++), we see the PINNACLE algorithm as being centered around the proposed convergence degree criterion and evolving eNTK eigenfunctions. Hence, we believe that our framework could be readily extended to other point selection heuristics that also select points with high convergence degrees (as per our proposed criterion) while also achieving some level of points diversity. This includes potentially weighted space-filling designs, as you have kindly suggested.
>
> We hope that our current framework will inspire future works on joint collocation and experimental points selection for PINNs.
>
> ---
>
> Thank you again for your insightful comments. We hope that we have sufficiently addressed your queries, and that our response will help strengthen your support for our paper.

---

> ### Author Response · Authors · 2023-11-22
> **Gentle reminder for Reviewer nqGc**
>
> As the rebuttal is coming to a close, we would like to thank you once again for your review of our paper, and hope that we have provided adequate clarifications to your questions. We would be happy to provide further responses otherwise.

---

### Official Review · Reviewer_WCRx · 2023-10-29

**Soundness:** 4 excellent
**Presentation:** 4 excellent
**Contribution:** 4 excellent
**Rating:** 8
**Confidence:** 4

**Summary:**

This paper proposes a method for adaptively selecting collocation points, which is necessary for training physics-informed neural networks. In particular, by treating the points related to experiments, partial-differential-equation constraints, and initial-boundary-value constraints in a unified manner, the proposed method automatically adjusts what proportion of collocation points should be allocated to each of them. The proposed method is based on a newly introduced criterion which is derived by using the theory of neural tangent kernels. The relationship between this criterion and generalization error has also been shown.

**Strengths:**

The choice of collocation points for PINNs is an important issue that has a significant impact on performance. In particular, this paper deals with collocation points in a unified way, independent of the type of loss functions associated with them. As far as I know, this is certainly a new approach and seems to be very promising.

In addition, the newly introduced criterion is derived with a theoretical basis and is highly reliable. This is just my impression but I suppose that the theorem that shows the relationship with the generalization error is itself valuable.

**Weaknesses:**

The strength of this paper seems to be that collocation points for initial boundary values and those for PDEs can be treated in a unified way, but a method of training networks without collocation points for initial boundary values is also proposed. When such a method is employed, the proposed method may lose a certain extent of significance.

**Questions:**

As mentioned above, for the initial boundary values, a method of designing neural networks to satisfy them has been proposed, e.g. in the following paper.

Lagaris, I. E., Likas, A., and Fotiadis, D. I. (1998). Artificial neural networks for solving ordinary and partial differential equations. IEEE Transactions on Neural Networks, 9(5):987–1000.

What is the significance of the proposed method when such a method is employed?

---

> ### Author Response · Authors · 2023-11-17
> **Response to Reviewer WCRx (Part 1/2)**
>
> We would like to thank the reviewer for taking the time to review our work, and we greatly appreciate the positive and insightful comments for our paper. We have provide clarifications to your comments and questions below.
>
> ---
>
> > The strength of this paper seems to be that collocation points for initial boundary values and those for PDEs can be treated in a unified way, but a method of training networks without collocation points for initial boundary values is also proposed. When such a method is employed, the proposed method may lose a certain extent of significance.
>
> > As mentioned above, for the initial boundary values, a method of designing neural networks to satisfy them has been proposed, e.g. in the following paper.
> > Lagaris, I. E., Likas, A., and Fotiadis, D. I. (1998). Artificial neural networks for solving ordinary and partial differential equations. IEEE Transactions on Neural Networks, 9(5):987–1000.
> > What is the significance of the proposed method when such a method is employed?
>
>
> While one of PINNACLE's strength lies in its ability to consider PDE and all initial/boundary conditions (IC/BC) collocation points in a unified way as you had kindly pointed out, we would like to emphasize that PINNACLE's strength also lies in:
>
> 1. **Better individually optimize each point type (e.g., PDE collocation points even when there are no IC/BC points)** based on the PINNs training dynamics as described in Sec. 4.2. To further demonstrate this, **we ran a simple example based on the method by Lagaris et al. that you have kindly raised, and showed that PINNACLE still outperform benchmarks even when considering just one training point type.**
>
> 2. **Its ability to jointly optimize experimental and just PDE collocation training points** as well in the absence of IC/BC points, which can be seen from the inverse experiment in our paper and also an additional experiment that we have run.
>
> For (1), this means that **even if there are only PDE collocation points to be considered, i.e., PDE forward problems with hard IC/BC constraints, PINNACLE's convergence degree criterion still applies** which could take into account the neural network's training dynamics as characterized by its Neural Tangent Kernel (NTK). The choice of PDE collocation points could still be optimized based on the criterion to focus on points to potentially increase training convergence and reduce generalization loss.
>
> - **We could see some signs of this optimized selection of PDE collocation points**, by observing the selected PDE collocation points by PINNACLE for the 1D Advection and 1D Burgers' problems (Fig. 3 of main paper). Apart from the shift in budget allocation of between IC/BC and PDE collocation points, **notice that the choice of PDE collocation points clusters around more informative regions**, such as the stripes in the Advection equation problem and areas with richer features in the Burgers' equation problem.
>
> - In addition, we have also performed a simple **demonstration involving only PDE collocation points, where we used the method by Lagaris et al.** that you have kindly raised, and showed that PINNACLE still outperforms benchmarks. We consider a simple 1D Diffusion equation forward problem where we use a PINN with hard constraints, with results in Fig. 23 of the updated Appendix. We can see that **even with just one type of training point (PDE collocation point), PINNACLE still provides a substantial training performance gain over other training point selection methods**. While the results are still preliminary, this shows that PINNACLE is likely useful even when we do not have to consider a joint training dynamics of multiple training point types.
>
> - We would like to also point out that while such methods that incorporate hard constraints are useful for certain settings, **it is still in general challenging to achieve such hard constraints for a wide range of other boundary conditions**. In those cases, PINNACLE would not only contribute in better PDE collocation point selection, but also joint optimization of all training point types.
>
> ---
>
> &#8595; **Continued below** &#8595;

---

> ### Author Response · Authors · 2023-11-17
> **Response to Reviewer WCRx (Part 2/2)**
>
> ---
>
> &#8593; **Continued from above** &#8593;
>
> ---
>
> For (2), this means that **even in situations where no IC/BCs were provided but experimental and PDE collocation points need to be selected, PINNACLE would still be able to provide significant value and outperform benchmarks**.
>
> - For example, we can see this in the 2D Navier-Stokes inverse problem setting (details in Appendix J.1, empirical results in Fig. 5 of our main paper), where no IC/BCs are imposed. In that example, we have two coupled PDEs involving 3 outputs each with 2+1 (time) dimensions, and our task is to estimate two unknown PDE parameters given experimental data on 2 out of the 3 outputs and no IC/BC conditions. We found that **PINNACLE was still able to outperform benchmarks in estimating the two unknown parameters, given its optimized joint selection of experimental and PDE collocation points alone**.
>
> - In addition, we have also performed an additional inverse experiment (added in Fig. 19 of the updated paper Appendix), **the 3D Eikonal inverse problem, where we tackle the challenging task of learning unknown inverse quantity that is a 3D field instead of just a few scalar PDE parameter values**. The Eikonal equation links wave propagation dynamics with the wave speed at different locations,and our task is to reconstruct the wave speed at each location which can be interpreted as representing the composition of the medium.  In this inverse problem, there are no IC/BCs, and we are able to query for experimental data regarding the time for wave to propagate to a certain 3D location. In this case, we see that **PINNACLE is also able to more efficiently select both experimental points and collocation points to achieve a substantial improvement in the inverse field reconstruction**.
>
> We thank the reviewer for suggesting this interesting PINN training setting which we can investigate further in future works.
>
> ---
>
> We hope that we have sufficiently addressed your queries, and that our response will help strengthen your support for our paper.

---

> ### Author Response · Authors · 2023-11-22
> **Gentle reminder for Reviewer WCRx**
>
> As the rebuttal is coming to a close, we would like to thank you once again for your review of our paper, and hope that we have provided adequate clarifications to your questions. We would be happy to provide further responses otherwise.

---

> > ### Comment · Reviewer_WCRx · 2023-11-22
> >
> > Thank you very much for the detailed response. Although I have not changed the score because it is already high, all my concerns have been addressed.

---

### Official Review · Reviewer_Eh91 · 2023-10-30

**Soundness:** 3 good
**Presentation:** 3 good
**Contribution:** 3 good
**Rating:** 8
**Confidence:** 3

**Summary:**

This paper proposes the PINNACLE, which is based on PINN network to address PDE problem. Specifically, proposed algorithm jointly optimizes the selection of all training points type and the authors demonstrate theoretically that criterion used by PINNACLE is related to generalization error of PINN.

**Strengths:**

1. The theoretical results are interesting. The author connects the proposed convergence degree notion with the generalization error bound.  and demonstrates how to approximate the optimal set for convergence degree.

2. The experimental results are great. Compared with other baselines, proposed methods introduce observable improvements.

**Weaknesses:**

1. For the K-MEANS++ method, the authors are encouraged to provide the time consumption comparison with other baselines. I am not sure if K-MENAS++ will introduce much extra computation cost.

2. For the K-MEANS++ method, the author claims that "this method select points with high convergence degrees". The authors are expected to provide more explanation why this method increase the convergence degree. It is the same for SAMPLING method. The authors are expected to explain how to "select a point which is proportional to $\hat{\alpha}(z)$ and how it improves the convergence degree" explicitly.

3. For the experiments, the authors are encouraged to provide the results mean error with running time rather than only steps. Furthermore, since PINNACLE achieves great results in 1D Advection and 1D Burger settings, it will be interesting to explore with more settings, e.g., 1D Wave equation and 1D KdV equation.

**Questions:**

Check the weakness.

---

> ### Author Response · Authors · 2023-11-17
> **Response to Reviewer Eh91 (Part 1/3)**
>
> We would like to thank the reviewer for taking the time to review our work, and we appreciate the positive comments regarding both the theoretical and experimental results. We have provided clarifications to your comments and questions below.
>
> ---
>
> > For the K-MEANS++ method, the authors are encouraged to provide the time consumption comparison with other baselines. I am not sure if K-MENAS++ will introduce much extra computation cost.
>
> > For the experiments, the authors are encouraged to provide the results mean error with running time rather than only steps.
>
> From our experiments, we find that **PINNACLE-K (the K-MEANS++ variant) achieves lower overall runtime compared to benchmarks even though the K-MEANS++ mechanism may incur computation costs for point selection**. This is because the better training points selected using this mechanism allows the model to converge faster and incur lower training time.
>
> - In Sec. 6.3. of the main paper, we mentioned that **PINNACLE is able to reach the target loss faster and achieve shorter computation time compared to other algorithms in experiments of forward and inverse settings**, though due to space constraints we only provided the supporting empirical results in the Appendix (Fig. 20 of the updated Appendix), which shows runtime results of the 1D Advection forward problem (Fig. 20a) and the 2D Navier-Stokes inverse problem (Fig. 20c). The runtime in these empirical results captures all computational costs of the algorithms, which consist of both training set selection time (including K-MEANS++ computational overheads) and also the model training time.
> - To better address your concern, we also ran additional timing experiments for a more challenging problem setting, the **3D Eikonal inverse problem where the unknown PDE parameter to be estimated is a 3D field rather than just a scalar quantity** (added to Fig. 20b). We demonstrate that even for this challenging problem setting, PINNACLE-K is also able to achieve much lower loss with a shorter amount of runtime. Hence, even from a runtime perspective there is strong motivation to use PINNACLE-K over other point selection mechanism even though the K-MEANS++ mechanism may incur additional computation costs.
>
> Additionally, we would like to stress that **the benefits from using PINNACLE significantly extends beyond reduced training time, especially in problems involving experimental data that is costly to obtain**. More specifically, in inverse problems where the goal is to learn unknown PDE parameters from experimental data (e.g. the 2D Navier-Stokes and 3D Eikonal settings mentioned above), the cost of acquiring these experimental data (e.g. from expensive experimental set-ups and measurements) is often much larger than PINN training costs. **PINNACLE is able to more efficiently select experimental data that is most beneficial to tackling the inverse problem, by taking into account overall PINNs training dynamics with optimized collocation point selection**.
>
> As other PINNs work on collocation points selection mainly report test error with respect to the number of training iterations, we have similarly done so in order to be consistent with them [1, 2].  However, we agree with the reviewer that the running time is also an important consideration in some problem setups, and will look to more prominently highlight some of our runtime results as space permits.
>
> ---
>
> #### References
>
> [1] Wu et al. A comprehensive study of non-adaptive and residual-based adaptive sampling for physics-informed neural networks. Computational Methods in Applied Mechanics and Engineering, 2023.
>
> [2] Krishnapriyan et al. Characterizing possible failure modes in physics-informed neural networks. NeurIPS 2021.
>
> ---
>
> &#8595; **Continued below** &#8595;

---

> ### Author Response · Authors · 2023-11-17
> **Response to Reviewer Eh91 (Part 2/3)**
>
> > For the K-MEANS++ method, the author claims that "this method select points with high convergence degrees". The authors are expected to provide more explanation why this method increase the convergence degree. It is the same for SAMPLING method. The authors are expected to explain how to "select a point which is proportional to and how it improves the convergence degree" explicitly.
>
> The SAMPLING and K-MEANS++ approaches that we proposed are heuristics meant to select points with high convergence degrees while also promoting some diversity of points selected, which is useful in the earlier stages of training as the eNTK would still be evolving (as we have discussed in Sec. 4.2).
>
> We first explain why the K-MEANS++ method selects points with high convergence degrees as claimed. This method, as noted in Sec. 5.2 of our paper, has also been used by active learning works to select points with large magnitudes in the embedding space while encouraging diversity of points [3]. To aid our explanation of the point selection mechanism, we have added additional plots in Fig. 9 in the updated Appendix.
>
> - For our K-MEANS++ method, we first represent each point with a vector embedding, where **its Euclidean norm will correspond to the convergence degree $\hat{\alpha}(z)$**, and hence points that have the largest Euclidean norm will have the largest individual convergence degree. We then perform the standard K-Means++ initialization on these embedded vectors to select a batch of points, which basically involves sequential sampling of $k$ points with the probability proportional to each point's Euclidean distance to the closest point that had already been selected.
>
> - For illustration, we can consider the empirical point distribution from one of our experiments (1D Burger's), visualized in a few embedding dimensions as blue points in Fig. 9a. We can make two observations: (a) **the points with low convergence degree (i.e., lower norm in embedded space) will all be "clustered" together close to the origin** and hence will be less likely to be selected by K-MEANS++ which preferentially samples points that are further spread out; (b) **the dimensions in the embedding space that correspond to the dominant eNTK eigenfunctions (higher $\hat{\lambda}$) will be stretched out**, leading to coefficients that are more spread out and cause points with high convergence degree to be further away from other points.
>
> - Fig. 9a highlights points that are eventually selected by PINNACLE-K in orange. Notice that (a) and (b) described above can be observed in the plot: **points with high convergence degrees are more widely spread out and are therefore preferentially selected**. Fig. 9b explicitly sorts the candidate points based on their convergence degrees, and further shows that PINNACLE-K mainly selects points that have much higher convergence degrees.
>
> For our SAMPLING method, we basically sample points with probability proportional to their convergence degrees $\hat{\alpha}(z)$. Hence, by design, points with larger convergence degree $\hat{\alpha}(z)$ will have a larger chance of being chosen by PINNACLE-S.
>
> ---
>
> #### References
>
> [3] Ash et al., Deep Batch Active Learning by Diverse, Uncertain Gradient Lower Bounds. ICLR 2020.
>
> ---
>
> &#8595; **Continued below** &#8595;

---

> ### Author Response · Authors · 2023-11-17
> **Response to Reviewer Eh91 (Part 3/3)**
>
> > Furthermore, since PINNACLE achieves great results in 1D Advection and 1D Burger settings, it will be interesting to explore with more settings, e.g., 1D Wave equation and 1D KdV equation.
>
> In the main paper, **we had provided some experiments involving more complex PDE and problem settings** beyond the 1D Advection and 1D Burgers time-dependent forward problem setting. In particular:
>
> - **We had run the 2D Navier-Stokes time dependent inverse problem setting**, which consisted of coupled PDEs involving 3 outputs, and there were also two unknown parameters that we had to estimate. We demonstrated that even in this challenging setting, PINNACLE outperformed benchmarks (Fig. 5) in learning the unknown parameters, and also managed to achieve a shorter computational time compared to benchmarks to achieve comparable loss (Fig. 17 in Appendix K.3).
>
> - We also considered **a challenging transfer learning setting for the 1D Advection PDE**, where we had to use a much tighter points budget (5 times fewer points) to fine-tune a pre-trained PINN that encoded the solution for a different initial condition (IC). For this setting, we also showed (in Fig. 6) that PINNACLE achieved significantly better performance than benchmarks.
>
> Nonetheless, to further demonstrate the effectiveness of PINNACLE, we **provided additional experiments in a wider range of challenging settings**. We summarize the additional problem settings and empirical results here, while the plots and additional details are in the updated paper appendix:
>
> - **The 1D KdV equation forward problem** (Fig. 17a), as you had kindly suggested. We show that PINNACLE has similarly outperformed benchmarks.
>
> - **The 2D Shallow Water forward problem** (Fig. 17b). This is a variant of the Navier-Stokes equation for describing the behavior of shallow water flows in two spatial dimensions and one temporal dimension, and consists of a **system of three coupled PDEs with four outputs**. In this case, we also find that PINNACLE is able to better predict the value of water depth compared to other methods.
>
> - **The 3D Eikonal equation inverse problem** (Fig. 19). The Eikonal equation links wave propagation dynamics with the wave speed at different locations. In this inverse problem, we are able to query for data regarding the time for wave to propagate to a certain 3D location, and our task is to reconstruct the wave speed at each location which can be interpreted as representing the composition of the medium. Notice that this is a challenging problem, as it deals with **3D input space and the task is to learn an unknown PDE parameter that is a 3D field rather than just scalar quantities**. In this case, we see that **PINNACLE is able to more efficiently select both experimental points and collocation points to achieve a substantial improvement in the inverse field reconstruction, while taking lower overall runtime**.
>
> These additional results provide further empirical support that PINNACLE is able to outperform benchmarks in a wide range of challenging settings, for both forward and inverse problems. We thank the reviewer for your suggestion, and we will look to include the results into the paper as space permits.
>
> ---
>
> Thank you once again for your comments. We hope our additional clarifications and responses have adequately addressed your concerns, and would improve your evaluation of our paper.

---

> > ### Comment · Reviewer_Eh91 · 2023-11-20
> >
> > Thanks for your detailed response. It addressed the experimental concern for me and I raise the score to 8.

---

> > > ### Author Response · Authors · 2023-11-22
> > >
> > > Dear Reviewer Eh91, we would like to thank you once again for your review, and for improving your evaluation of our paper in response to our rebuttal.

---

### Official Review · Reviewer_hySh · 2023-11-05

**Soundness:** 3 good
**Presentation:** 3 good
**Contribution:** 3 good
**Rating:** 6
**Confidence:** 3

**Summary:**

Joint selection of experimental points and collocation points in PINN training is proposed in this paper to achieve better training results for PINNs. A point selection algorithm that aims to select a training set that maximizes the convergence degree is proposed, which can lead to lower generalization bounds.

**Strengths:**

The paper contains a substantial amount of mathematical proofs. The experimental results are excellent, and the algorithm exhibits significantly higher accuracy compared to other algorithms.

**Weaknesses:**

The algorithm seems straightforward, and the presentation can be more concise.

**Questions:**

1. In Algorithm 1, how to select SAMPLING or K-Means sampling strategy?
2. Is the calculation of NTK used to compute the convergence degree?
3.The runtime comparison is not provided.

---

> ### Author Response · Authors · 2023-11-17
> **Response to Reviewer hySh (Part 1/2)**
>
> We would like to thank the reviewer for taking the time to review our work, and for recognizing that our work is backed by a "substantial amount of mathematical proofs" and that "the experimental results are excellent".
> We have provided clarifications to your questions and concerns below.
>
> ---
>
> > The algorithm seems straightforward, and the presentation can be more concise.
>
> We would like to emphasize that while our novel algorithm (PINNACLE) may seem straightforward at first glance, **the analysis and insights leading to the algorithm are challenging and non-trivial, and may themselves also be of broader interest** to the PINNs community. These all build up to PINNACLE, which we aimed for it to be **simple in form for straightforward implementation, but also theoretically-motivated and with strong empirical performance**, as you have kindly pointed out.
>
> Our paper could have been made more concise if we were to focus primarily on the algorithm and empirical results, but we decided to **include additional insights and elaboration where relevant to help provide a better understanding of phenomena and design choices underlying our algorithm** for readers who may find them useful.
> Here, we highlight key challenges and some of the contributions we have addressed in our work:
>
> - *How could selecting points intelligently improve PINNs' training?* Past works have predominantly focused on optimizing just the selection of PDE collocation points to try improve PINNs performance, which may naturally seem to be the only consequential point type for learning PDE solutions. It was non-trivial to go beyond the prevailing norm and realize that the various terms in the PINN composite loss function (Eq. 2 in the paper) **corresponding to different point types (experimental points, PDE and IC/BC collocation points) are actually tightly coupled** with each other, leading to training dynamics that is complex but **if understood could be exploited to significantly boost training performance**.
>     - We are the first to adopt this perspective and **provide a unifying treatment for all types of training collocation and experimental points through an augmented space** (Sec. 4.1) that allows them to be considered jointly.
>
> - *How can we analyze the PINN training dynamics involving all experimental and collocation point types?* The cross interactions among the various point types are complex and trying to exploit or even analyze them may seem an intractable challenge.
>     - Based on the augmented space view, we managed to **both theoretically and empirically analyze the PINN training dynamics** using the combined PINN empirical Neural Tangent Kernel (eNTK) and its eigenfunctions (Sec. 4.2).
>     - We also explicitly considered the **PINN eNTK evolution during training**, unlike many past NTK works that assumes a fixed NTK, and demonstrated how the eigenfunctions would evolve such that the dominant ones would be able to reconstruct the final PINN solution.
>     - Our analysis of PINNs training dynamics using the combined eNTK eigenfunctions, to the best of our knowledge, have not been considered in previous PINN works, and **may be of broader interest** beyond its use in the development of the PINNACLE algorithm.
>
> - *How can we design a criterion that leads to faster training convergence and lower generalization error loss?* Even if there were exploitable cross-information among point types to achieve faster training convergence, it would be non-trivial to design a criterion for point selection that would also lead to lower generalization error loss.
>     - We proposed a **novel criterion, the convergence degree**, based on our analysis of the training dynamics (Sec. 4.3), that is also **theoretically shown to be related to lower generalization error loss**.
>
> - *How do we come up with an effective and efficient algorithm that uses our theoretically-motivated criterion to achieve significant performance improvement empirically?* It is often challenging to convert a theoretically-motivated approach to a practical algorithm that is straightforward to effectively implement.
>     - To achieve this, we proposed a method of **approximating the proposed criterion using the Nystrom approximation** (Sec. 5.1). We find that the method of approximating the true convergence degree criterion through approximation of eNTK eigenfunctions involves kernel methods which has not been previously used in training point selection for PINNs.
>     - Along with additional design choices, the resulting PINNACLE algorithm is theoretically-motivated while still being relatively **straightforward and elegant, and is able to provide strong empirical performances** for a wide range of problem settings, including forward, inverse and transfer learning problems.
>
> ---
>
> &#8595; **Continued below** &#8595;

---

> ### Author Response · Authors · 2023-11-17
> **Response to Reviewer hySh (Part 2/2)**
>
> >  In Algorithm 1, how to select SAMPLING or K-Means sampling strategy?
>
> In general, we suggest first trying PINNACLE-K (K-MEANS++) over PINNACLE-S (SAMPLING). Empirically, we found that **both PINNACLE-K and PINNACLE-S methods tend to produce similar results, though in some cases PINNACLE-K may be able to produce lower loss faster at early stages of training**. This may be because PINNACLE-K places a greater emphasis on point diversity during selection, which could be more important earlier in training when the eNTK tends to evolve more (Sec. 4.2), before it stabilizes nearer the end of training. PINNACLE-S, on the other hand, is better motivated theoretically as it involves sampling from an implicit distribution (whose p.d.f. is proportional to $\hat{\alpha}$), and therefore is more directly compatible with assumptions required in Theorem 1 on generalization error.
>
> ---
>
> > Is the calculation of eNTK used to compute the convergence degree?
>
> The short answer is yes. The convergence degree $\alpha$  (Eq. 10) is defined based on the eigenfunctions and eigenvalues of the PINN eNTK. The eigenfunctions and eigenvalues are empirically estimated using the Nystrom approximation, which requires the computation of the eNTK of a small reference set $Z_{ref}$, as shown in Eq. 14. We also show, in Proposition 1, that our use of Nystrom approximation with the eNTK provides a good estimation of the true convergence degree criterion.
>
> ---
>
> > The runtime comparison is not provided.
>
> In Sec. 6.3. of the main paper, we mentioned that **PINNACLE is able to reach the target loss faster and achieve shorter computation time compared to other algorithms in experiments of forward and inverse settings**, though due to space constraints we only provided the supporting empirical results in the Appendix (Fig. 20 of the updated Appendix), which shows **runtime results of the 1D Advection forward problem and the 2D Navier-Stokes inverse problem**.
>
> The runtime in these empirical results captures all computational costs of the algorithms, which consist of both training set selection time and also the model training time. The results showed that **the benefits gained from faster training convergence and better generalization error due to better point selection outweighs the additional overheads from PINNACLE's point selection**.
>
> However, to better address concerns about running time, we also ran **additional timing experiments for a more challenging problem setting, the 3D Eikonal inverse problem where the unknown PDE parameter to be estimated is a 3D field rather than just a scalar quantity**. The additional plot can be found in Fig. 20b in the Appendix. We find that even for this challenging problem setting, PINNACLE is able to achieve much lower loss in a shorter amount of runtime.
>
> Overall, we can see that PINNACLE **select points that can result in faster training convergence, and hence lower overall runtime despite incurring some computation time to select the points**. We find that this gives a strong motivation to use PINNACLE over other point selection mechanism even though the point selection process itself may incur additional computational costs.
>
> Additionally, we would like to highlight that **PINNACLE is also able to provide significant advantages beyond reducing training time, particularly in problems of selecting costly experimental data**. For example, in inverse problems (such as the 2D Navier-Stokes and 3D Eikonal examples) where the goal is to learn unknown PDE parameters from experimental data, the cost of acquiring these experimental data is often much larger than PINN training costs, due to the need of performing physical experiments and measurements to obtain these data. PINNACLE is able to more efficiently select experimental data that is most beneficial to tackling the inverse problem, by taking into account overall PINNs training dynamics with optimized collocation point selection.
>
> We will work towards more prominently highlighting these discussions/additional results as space permits.
>
> ---
>
> Thank you once again for reviewing our work. We hope that our responses and clarifications will improve your evaluation of our paper.

---

> > ### Author Response · Authors · 2023-11-22
> > **Gentle reminder for Reviewer hySh**
> >
> > As the rebuttal is coming to a close, we would like to provide a gentle reminder that we have posted a response to your comments. May we please check if our responses have addressed your concerns and improved your evaluation of our paper?

---

> ### Author Response · Authors · 2023-11-23
> **Reminder for Reviewer hySh**
>
> Dear Reviewer hySh,
>
> As we are less than 8 hours to the end of the rebuttal, may we please check if our responses have addressed your concerns and improved your evaluation of our paper?
>
> We are happy to provide further clarifications to address any other concerns that you may still have before the end of the rebuttal.
>
> Thank you very much.

---

> > ### Comment · Reviewer_hySh · 2023-11-23
> >
> > I have no further concerns. I will change my scores according to your feedback.

---

### Meta-Review · Area_Chair_qfc5 · 2023-12-03

**Metareview:**

This work introduces PINNACLE, a novel algorithm for training Physics-Informed Neural Networks (PINNs) that optimally selects collocation and experimental points by jointly considering their interactions and dynamically adjusting proportions during training, demonstrating better performance compared to existing point selection methods.

The reviewers think the paper makes some substantial novel contributions, both in terms and theory and also in terms of practical improvements over existing baselines. Some minor concerns initially raised by the reviewers were addressed by the authors in the rebuttal. I therefore recommend acceptance.

**Justification For Why Not Higher Score:**

The theory part of the paper is based on standard arguments based on the NTK. I think this could still be oral, but my personal view is that it would fit more as a spotlight (due to rather standard theoretical arguments).

**Justification For Why Not Lower Score:**

Significant empirical results.

---

### Decision · Program_Chairs · 2024-01-16

Accept (spotlight)